# Structural basis of membrane recognition of *Toxoplasma gondii* vacuole by Irgb6

Yumiko Saijo-Hamano[1] , Aalaa Alrahman Sherif[2], Ariel Pradipta[3,4], Miwa Sasai[3,4,5], Naoki Sakai[6] , Yoshiaki Sakihama[1], Masahiro Yamamoto[3,4,5] , Daron M Standley[2] , Ryo Nitta[1]

The p47 immunity-related GTPase (IRG) Irgb6 plays a pioneering role in host defense against *Toxoplasma gondii* infection. Irgb6 is recruited to the parasitophorous vacuole membrane (PVM) formed by *T. gondii* and disrupts it. Despite the importance of this process, the molecular mechanisms accounting for PVM recognition by Irgb6 remain elusive because of lack of structural information on Irgb6. Here we report the crystal structures of mouse Irgb6 in the GTP-bound and nucleotide-free forms. Irgb6 exhibits a similar overall architecture to other IRGs in which GTP binding induces conformational changes in both the dimerization interface and the membrane-binding interface. The membrane-binding interface of Irgb6 assumes a unique conformation, composed of N- and C-terminal helical regions forming a phospholipid binding site. In silico docking of phospholipids further revealed membrane-binding residues that were validated through mutagenesis and cell-based assays. Collectively, these data demonstrate a novel structural basis for Irgb6 to recognize *T. gondii* PVM in a manner distinct from other IRGs.

## Introduction

Infection by intracellular pathogens stimulates innate and acquired immune systems to produce IFN. IFN-γ is a proinflammatory cytokine produced from natural killer cells and T cells. Binding of IFN-γ with IFN-γ receptors activates gene expression programs via the JAK-STAT pathways. A number of IFN-γ–inducible products play pivotal and pleiotropic roles in cell-autonomous immunity against various intracellular pathogens such as viruses, bacteria and protozoan parasites (MacMicking, 2012).

*Toxoplasma gondii* is an important human and animal pathogen that causes lethal toxoplamosis in immune-compromised individuals such as those receiving bone marrow transplantations or suffering from AIDS (Goldstein et al, 2008; Boothroyd, 2009). IFN-γ

suppresses intracellular *T. gondii* growth in a manner dependent on inducible nitric oxide production by nitric oxide synthase 2 and tryptophan degradation by indoleamine 2,3-deoxygenase (IDO), both of which are important for prevention of chronic toxoplasmosis (Scharton-Kersten et al, 1997; Divanovic et al, 2012; Sasai et al, 2018). In contrast, recent studies demonstrate that host defense during acute toxoplamosis requires IFN-γ–inducible GTPases that localize at a *T. gondii*-forming vacuole called the parasitophorous vacuole (PV) inside infected cells and destroy the structure, leading to parasite killing. IFN-γ–inducible GTPases involving anti–*T. gondii* cell-autonomous immunity consist of p47 immunity-related GTPases (IRGs) and p65 guanylate binding proteins (GBPs) (Howard et al, 2011; Yamamoto et al, 2012; Saeij & Frickel, 2017). Most IRGs and GBPs are recruited to PV membranes (PVMs) and cooperatively disrupt the membrane structure. Sequential and hierarchical recruitment of IRGs and GBPs leads to efficient PVM disruption and pathogen clearance (Khaminets et al, 2010). The IRG Irgb6 has been shown to be localized at the PVM soon after *T. gondii* invasion to host cells and acts as a pioneer for the recruitment of other IRGs and GBPs (Khaminets et al, 2010; Lee et al, 2020). Genetic ablation of Irgb6 results in severely impaired accumulation of other IRGs and GBPs, reflecting the pioneering role of Irgb6 to induce host defense (Lee et al, 2020).

Recent reports of crystal structures of Irga6 in various nucleotide states and Irgb10 in the GDP state elucidated the basic architecture of IRGs, consisting of a GTPase domain and N-terminal and C-terminal helical domains (Ghosh et al, 2004; Ha et al, 2021). Structural studies also indicated that homodimerization through the GTPase domain interface is required to activate the GTPase of IRG proteins (Pawlowski et al, 2011; Schulte et al, 2016; Ha et al, 2021). However, the structural mechanism of PVM recognition is still unclear. Irga6 and Irgb10 use a myristoylated glycine at their N-terminus to attach to the PVM (Haldar et al, 2013), although detailed knowledge of the N-terminal structure is missing because of its flexibility. Irgb6 does not have the myristoylated glycine and instead recognizes phospholipids such as phosphoinositide 5P (PI5P) and phosphatidylserine (PS) via the C-terminal amphipathic

[1]Division of Structural Medicine and Anatomy, Department of Physiology and Cell Biology, Kobe University Graduate School of Medicine, Kobe, Japan    [2]Depertment of Genome Informatics, Research Institute for Microbial Diseases, Osaka University, Osaka, Japan    [3]Department of Immunoparasitology, Research Institute for Microbial Diseases, Osaka University, Osaka, Japan    [4]Laboratory of Immunoparasitology, World Premier International Immunology Frontier Research Center, Osaka University, Osaka, Japan    [5]Division of Microbiology and Immunology, Center for Infectious Disease Education and Research, Osaka University, Osaka, Japan    [6]RIKEN SPring-8 Center, Hyogo, Japan

Correspondence: ryonitta@med.kobe-u.ac.jp

α-helices to bind present in the PVM (Lee et al, 2020). Because of the lack of structural information on Irgb6, however, the structural basis for such phospholipid recognition and its relationship with nucleotide binding remain unclear.

Here, we aimed to elucidate the PVM recognition mechanism of Irgb6 by X-ray crystallography. We further investigated the membrane-binding interface by in silico phospholipid docking, followed by validation using mutational analyses.

# Results

## Overall architecture of the Irgb6 monomer in two distinct nucleotide states

To explore the atomic structure of Irgb6, full-length mouse Irgb6 was expressed and purified. Size exclusion chromatography (SEC) of purified Irgb6 produced two peaks (Fig S1A). SDS–PAGE analysis indicated that Irgb6 was mainly eluted in the second peak. Considering that the estimated molecular weight of Irgb6 is 47.3 kD, the eluted Irgb6 in the second peak should be a monomer. We also examined the GTPase activity of purified Irgb6 through anion exchange chromatography, indicating retained GTPase activity of Irgb6 (Fig S1B).

We successfully crystallized the monomer fraction of Irgb6. As indicated by the SEC analysis, one molecule of Irgb6 was included in the unit cell. The atomic structures of mouse Irgb6 monomer were solved in two states—with GTP and without nucleotide (nucleotide free: NF)—at 1.5 and 2.0 Å resolution, respectively (Fig 1A and B and Table S1). The former structure possesses GTP in the nucleotide-binding pocket (GTP-bound Irgb6), though the phosphate part of GTP took on unusual twisted geometries and the location of the coordinated magnesium ion was not obvious (Fig S2A). This conformation might be caused by strain induced through excessive interactions of three phosphates with the G1/P-loop and G2/switch I residues (Fig S2B). The NF structure did not have any nucleotide or ion in the pocket (NF Irgb6) (Fig S2C).

The overall architectures of Irgb6 are similar to previously solved Irga6 or Irgb10 structures (Ghosh et al, 2004; Ha et al, 2021). They consist of an N-terminal helical domain (N-domain; amino acids 1–55; αA-αC) (purple in Fig 1A), a GTPase domain (G-domain; amino acids 56–239; H1-H5, αd, S1-S6) (green in Fig 1A), and a C-terminal helical domain (C-domain; amino acids 255–415; αF-αL) (pink in Fig 1A) (Fig S3A). Helix αE serves as a linker among the three domains (amino acids 240–254) (brown in Fig 1A). The G-domain of Irgb6 exhibits a dynamin-like α/β structure with a central β-sheet surrounded by helices on both sides. The N- and C-domains stand side by side and are composed of 11 helices, most of which align parallel or antiparallel.

## Nucleotide-dependent conformational change of Irgb6

The Irgb6 structures in two distinct nucleotide states take on similar conformations with overall root-mean square deviation (RMSD) of 2.4 Å (Fig 1B). The N-domain is apparently in the same conformation, with an RMSD of 0.3 Å, demonstrating no conformational change

observed during GTP binding. On the other hand, the G- and C-domains change their conformation significantly with overall RMSDs of 2.8 and 2.2 Å, respectively.

The conformational changes of the G-domain are concentrated around the nucleotide-binding pocket, consisting of five consensus sequences among p47 GTPases (green in Fig 1C and blue box in Fig S3A). The G1/P-loop (GxxxxGKS) in the GTP-bound state recognizes α- and β-phosphates of GTP, whereas that in the NF state override the corresponding binding site of the α- and β-phosphates, kicking the GDP out from the nucleotide-binding pocket of Irgb6 (Figs 1D and S2A and C). The G2/switch I region follows the G1/P-loop and helix H1. Thus, the conformational change of G1/P-loop directly transduces changes in H1 and G2/switch I, although the G2 loop was almost invisible in both structures because G2 did not stably co-ordinate to the γ-phosphate (Figs 1C and D and S2A and C). Conformational changes were not apparent in the G3/switch II region (Fig 1D). Switches I and II are necessary for the hydrolysis of GTP by coordinating the γ-phosphate. This γ-phosphate recognizing reaction is referred to as an "isomerization" (Moore et al, 1993; Wittinghofer et al, 1997; Nitta et al, 2004, 2008). In both Irgb6 structures solved here, however, neither switches I nor II coordinate to the γ-phosphate of GTP; thus, our GTP-bound Irgb6 takes on a pre-isomerization state (Figs 1C and D and S2A). Apparent density corresponding to the $Mg^{2+}$ was not also observed even in the 1.5 Å resolution map (Fig S2A).

The G4 and G5 regions recognize the base of GTP (Fig 1C). The G4 and following helices αd and H4 change their conformation largely from the NF state. The αd of the GTP-bound state forms an α-helix by a loop-to-helix transition from the NF state, inducing a clockwise rotation of helix H4 (Fig 1D). It should be noted that, in the Irga6 or Irgb10 structure, these helices αd and H4 serve as an interface for homodimerization (Pawlowski et al, 2011; Schulte et al, 2016; Ha et al, 2021). Homodimerization is thought to be required to activate the GTPase of IRGs. By analogy with Irga6 and Irgb10, therefore, nucleotide-binding or release appear to initiate or break homo-dimerization of Irgb6, respectively, to control the GTPase activity of Irgb6.

A conformational change in the C-domain was observed around helices αH, αI, and αLb (Fig 1A and B). These helices do not contact directly to either the G-domain or the neighboring molecule in the crystal packing environment. Thus, how these conformational changes are induced by nucleotide binding is still not clarified.

Finally, we examined the inter-domain rearrangement from the NF state to the GTP-bound state by superimposing two structures on their G-domains. As a result, N- and C-domains cooperatively rotated 5° in a counterclockwise direction around the G-domain during GTP binding (Fig 1E). The linker helix αE plays a pivotal role to transduce this nucleotide-dependent conformational change. It makes contacts not only with the preceding loop to switch I (G1-G2 loop) through hydrogen bonds between the main chains and Gln251, but also with the S2-S3 loop preceding to the switch II region through the hydrophobic residues Leu106, Val109, Val245, and Leu248 (inset of Fig 1E). Therefore, the helix αE can sense the conformational change of two switch regions and transduce the change to the N- and C-domains. From the NF state to the GTP-bound state observed here, the conformational change of switch I pushes the αE toward the helical domains, generating rotational changes in the N- and C-domains (Fig 1E).

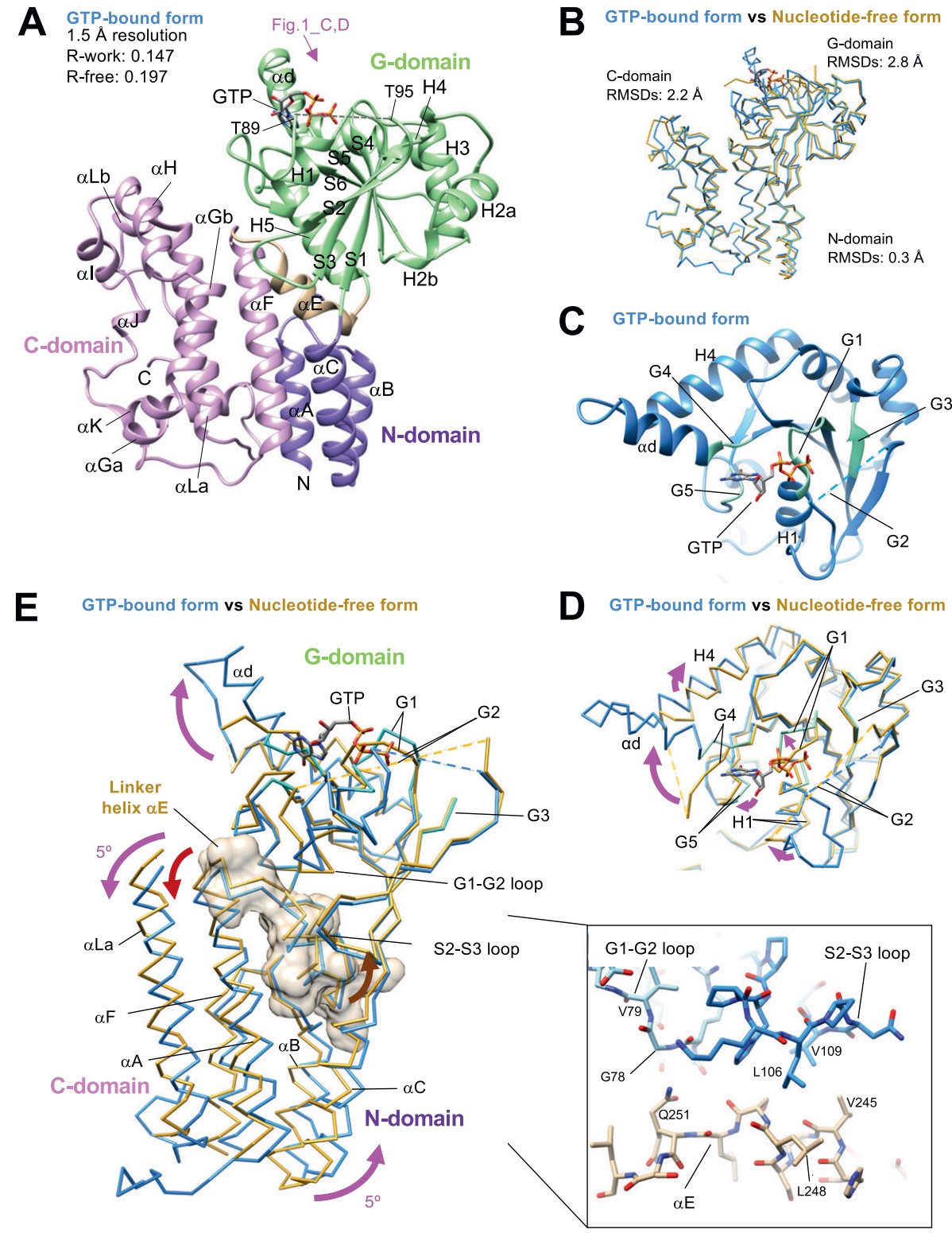

**Figure 1. Crystal structures of Irgb6 with GTP and without any nucleotide.**
**(A)** Crystal structure of Irgb6 with GTP. **(B)** Structural comparison between Irgb6 with GTP (blue) and without any nucleotide (yellow-brown), superimposed using all residues to minimize root-mean square deviations. **(A, C)** Structure around the nucleotide-binding pocket in G-domain observed from the top indicated by pink arrow in panel (A). **(D)** Conformational change of nucleotide-binding pocket during GTP binding. **(E)** Conformational change of N- and C-domains during GTP binding. Irgb6 with GTP (blue) and without any nucleotide (yellow-brown) was superimposed on their G-domains to illustrate the relay of conformational changes from the nucleotide-binding pocket. Whole N-domain, main components of G-domain around the nucleotide-binding pocket, linker helix αE, and helices αG and αLa of C-domain are shown. Linker helix αE is also shown with surface model. (Inset) Close-up view of the interactions between switches I–II and the helix αE.

## Comparison of Irgb6 structures with Irga6 and Irgb10 structures

We next compared the newly solved Irgb6 structures with previously solved IRG structures. Irga6 structures were reported in three states with GMPPNP (PDB ID: 1TQ2), GDP (PDB ID:1TPZ), and without any nucleotide (PDB ID: 1TQD) (Ghosh et al, 2004). The Irgb10 structure was reported in the GDP state (PDB ID: 7C3K) (Ha et al, 2021). We thus compared our Irgb6 structures with these four structures by superimposing them using each N-, G-, and C-domains (see the Materials and Methods section for detail). Fig 2A shows RMSDs of N-, G-, and C-domains among six structures. This comparison can be summarized in three main findings: (1) The N-domain assumes a very similar structure among IRGs except for the N-terminal end; (2) the structural similarities of the G-domain reflect the nucleotide state of IRGs; (3) the C-domain exhibits large structural variation among IRGs.

The N-domains of the six structures take on very similar conformations, with RMSDs less than 1.0 Å (Figs 2A and B and S3B). However, marked difference between Irgb6 and the others exists. The N-terminal helix αA of Irgb6 is slightly longer than those of Irga6 and Irgb10 (Figs 2B and S3B). Instead, Irga6 and Irgb10 have an ~15-residue addition before helix αA (Fig S3A). This includes the N-terminal glycine residue which is crucial for PVM localization of Irga6 and Irgb10. Gly2 is known to be myristoylated, which allows it to bind to the PV membrane. Because Irgb6 does not have this additional sequence or an equivalent glycine, a different mechanism for recruiting Irgb6 to the PVM exists, as detailed below.

The G-domain of the GTP-bound Irgb6 is most similar to the GDP form of Irga6 (RMSD = 1.6 Å) or Irgb10 (RMSD = 1.3 Å), rather than the GMPPNP form of Irga6 (RMSD = 2.2 Å) (Fig 2A and C). This tendency was similar when they were aligned using the G-domain without G2/G3 sequences. This is reasonable because the G1, G4, and G5 sequences recognize α-, β-phosphates and nucleotide base; whereas, γ-phosphate is not trapped by G2/G3 sequences (Figs 1C and S2A and B). In our GTP-bound structure, therefore, Irgb6 only recognizes a "GDP part" of GTP so that it resembles the GDP form of IRG proteins. By analogy with Irga6 or Irgb10 (Ghosh et al, 2004; Ha et al, 2021), homodimerization of Irbg6 might trigger the isomerization of the G-domain to assume the active GTP form.

The G-domain of NF Irgb6 exhibits a similar conformation to the NF form of Irga6, although the helix H4 and surrounding structures assume different conformations (Fig 2D). This difference can be explained by the increased flexibility of G4 because of the absence of nucleotide, which is further stabilized by a neighboring molecule in the crystal packing environment (Fig S3C). Therefore, the structure of the G-domain and its conformational changes during the GTPase cycle are basically conserved among IRGs.

## Unique conformation of C-domain in Irgb6 structures

In comparison with the N- and G-domains, the structural similarity of the C-domain is low among IRGs (Fig 3A). The RMSDs between different subfamilies of IRGs are greater than 3 Å (Fig 2A). The C-terminal 22 residues of the C-domain helix αLb and the following tail, are unique additions in Irgb6 (Fig S3A). The αLb helix is rich in basic residues, whereas the tail is rich in acidic residues. The biological significance of the C-terminal tail is currently unknown.

There are two antiparallel long helices, αF and αLa, which take on well conserved conformations among IRGs, penetrating the N- and C-domains (Fig 3A). Observed from the bottom side, the opposite face to the GTP binding interface, the helix pair is located at the center of the N- and C-domains, surrounded by five helices from the N- and C-domains (helices αA, αB, and αC in N-domain and helices αGa and αK in C-domain) (Fig 3B). Two connecting loops, the αF-αGa loop and αK-αLa loop, extend from the central helix pair. These loops, as well as the surrounding five helices, change the conformation significantly among IRGs (Fig 3C). The C-domain helices of Irgb6 rotate counterclockwise around the central pair, whereas the N-domain helices of Irgb6 rotate toward the clockwise direction, thus closing the cleft between N- and C-domains (dashed lines in Fig 3B and C). These helices are connected through hydrophobic contacts where residue Trp3 of the αA acts as a keystone (Fig 3B and D). Trp3 takes alternative conformations and links two connecting loops with three N-terminal helices, thus contributing to the interdomain contact between N- and C-domains. Also, the aromatic residues connect the helix αA with αF, supporting the cooperative movement of N- and C-domains (Fig 3D). The unique conformation of the αF-αGa loop in Irgb6 is also supported by the hydrophobic residues Phe350 and Ile353 of helix αK, which also assumes a unique conformation because of the long insertion of αK-αLa loop (Fig 3E).

Quite suggestively, two basic residues Lys275 and Arg371, which are necessary for PVM recruitment of Irgb6, are located at the ends of the central pair (Fig 3B) (Lee et al, 2020). Considering that the myristoylation site of Irga6 and Irgb10 exists at the N-terminal end, close to the end of central pair, these sites were assumed to contribute to the binding of Irgb6 to the PVM.

## Docking simulation of phospholipids to the Irgb6

We previously reported that Irgb6 binds to PI5P and PS, which are both components of the *T. gondii* PVM (Lee et al, 2020). Thus, we simulated the docking of various phospholipids to our Irgb6 structure to investigate the specificity. The αF-αGa loop was not visualized in the NF Irgb6 because of its high flexibility, whereas it was well defined in the GTP-bound Irgb6. We therefore used the GTP-bound Irgb6 for the docking experiments.

Molecular docking was performed to investigate the interaction between Irgb6 protein with four phospholipids (PI5P: PubChem 643966, PS: PubChem 9547090, PE: PubChem 445468 and PC: PubChem 160339) (Fig S4A) using Glide (Halgren et al, 2004). As a consequence, the head groups of phospholipids were docked on the αF-αGa loop and the central helix pair (Fig 4A and B). Hereafter, we thus denoted the αF-αGa loop as the "PVM-binding loop." To extend sampling of this region, the grid box was approximately centered on residues Trp3, Lys275, and Arg371, with small perturbations, and two rotamer states for Trp3 and Arg371 independently considered, for a total of six docking runs per ligand. Because glide measures the ligand–receptor binding free energy in terms of Glide Score, we compared all six Glide Scores of four phospholipids to evaluate their binding affinity to Irgb6. Consistent with our previous report (Lee et al, 2020), the lower mean Glide scores of the polar head groups indicated that the binding free energy of Irgb6 to the

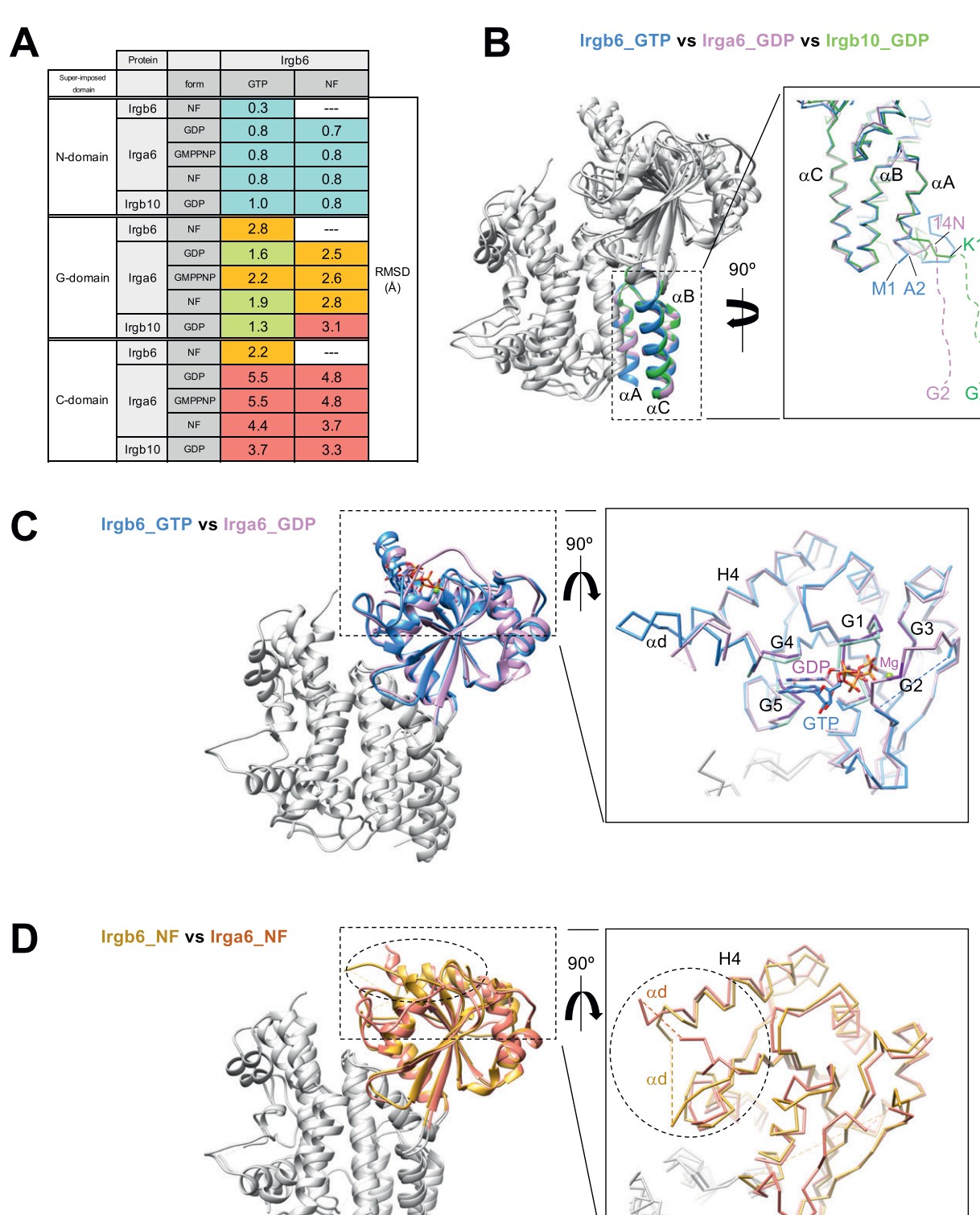

**Figure 2. Structural comparison among Irgb6, Irga6, and Irgb10.**
**(A)** Root-mean square deviations (RMSDs) among Irgb6, Irga6, and Irgb10, superimposed on their N-, G-, and C-domains. RMSDs≦1: cyan, 1 < RMSDs ≦ 2: yellow-green, 2 < RMSDs ≦ 3: orange, 3 < RMSDs: red. **(B)** Structural comparison among Irgb6 with GTP (blue), Irga6 with GDP (pink), and Irgb10 with GDP (green) superimposed on their N-domains. **(C)** Structural comparison between Irgb6 with GTP (blue) and Irga6 with GDP (pink), superimposed on their G-domains. **(D)** Structural comparison between Irgb6 without any nucleotide (yellow-brown) and Irga6 without any nucleotide (red), superimposed on their G-domains.

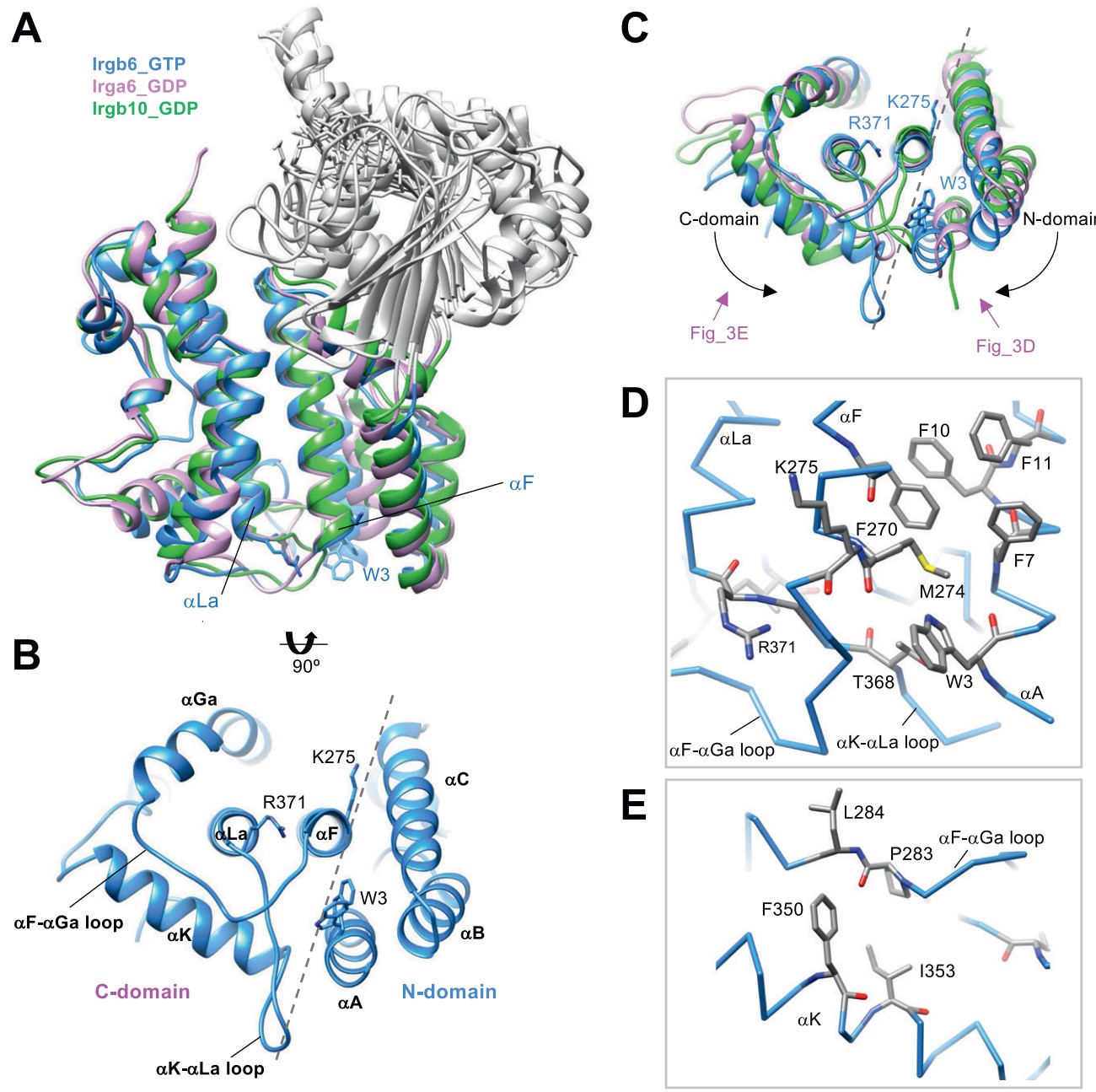

**Figure 3. Unique conformation of C-domain in Irgb6.**
**(A)** Structural comparison among Irgb6 with GTP (blue), Irga6 with GDP (pink), and Irgb10 with GDP (green) superimposed on their C-domains. **(B)** Bottom view of Irgb6. Broken line indicates the boundary between N- and C-domains. **(C)** Bottom view of panel (A) showing the conformational differences among Irgb6 with GTP (blue), Irga6 with GDP (pink), and Irgb10 with GDP (green). **(D)** Close-up view of the boundary indicates the interaction between the helices αA and αF. **(E)** Close-up view of the interaction between the αF-αGa loop and the helix αK.

PI5P polar head was lower than that of PS, PE, or PC (Figs 4C and S4B).

The tips of phosphate groups of PI5P bind directly to Arg371 by hydrogen bonds and form salt bridges and hydrogen bonds to Lys 275 via several water molecules (Fig 4A). The inositol makes hydrophobic contact with Leu279 of the PVM-binding loop. The following phosphate faces toward the N-terminal helices, thus the acyl chain would extend toward the N-terminal helices. This surface is mainly covered by hydrophobic residues (Trp3, Ile33, Leu38, and Val42), thus environmentally preferable to acyl chain extension (Fig 4A and D). The head group of PS partially extends toward the hydrophobic surface, whereas those of PC and PE only occupy the left side of the pocket (Fig 4B). These properties might reflect the preference of phospholipids by Irgb6.

Glide docking of PI5P was further confirmed by introducing mutations to Trp3 at the N-terminus (W3A) or to the PVM-binding

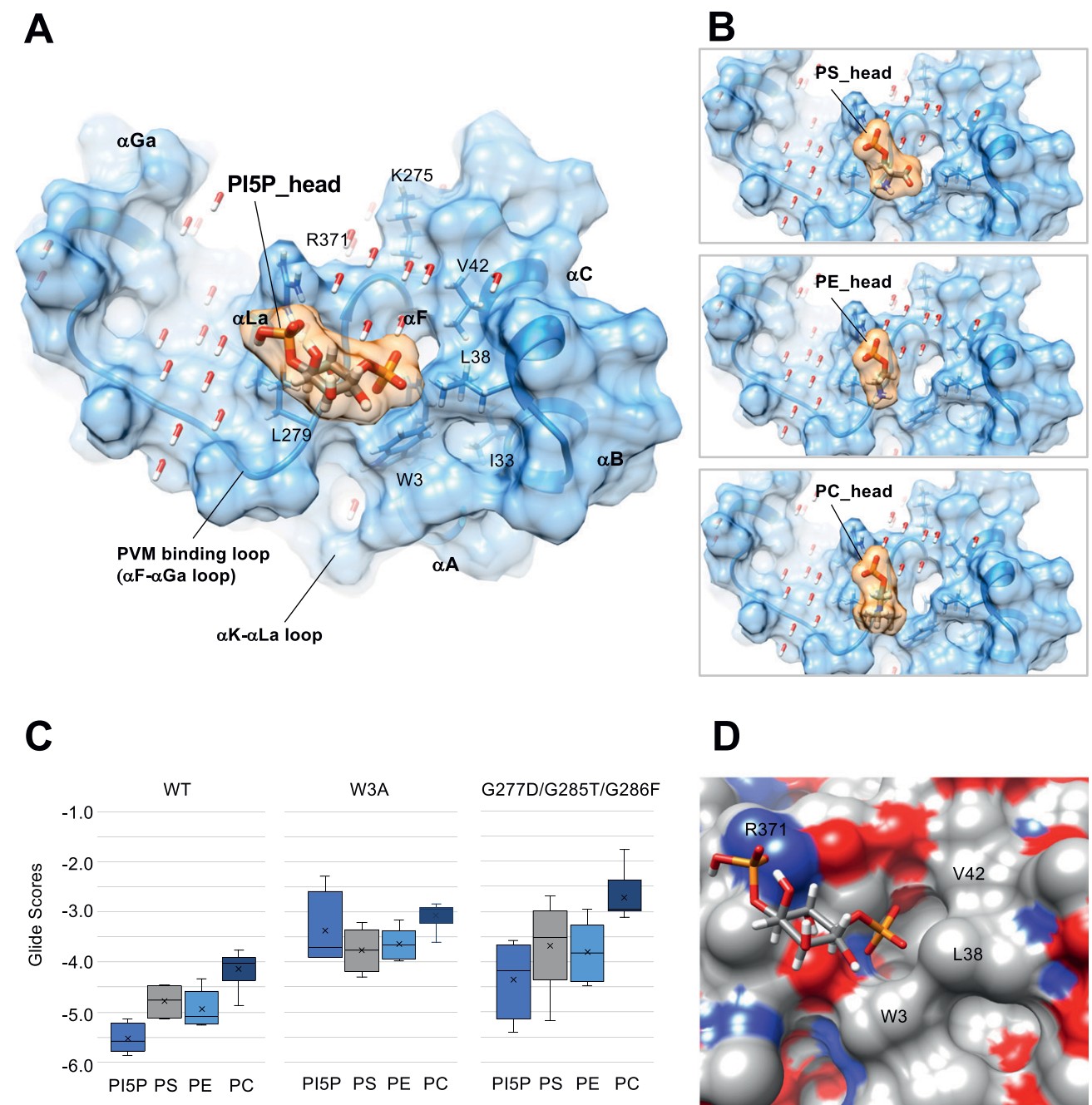

**Figure 4.  Docking simulation of phospholipids to the Irgb6.**
**(A)** Docking of the polar head of PI5P to the GTP-bound Irgb6. **(B)** Docking of the polar head of PS, PE, and PC to the GTP-bound Irgb6. **(C)** Glide scores of Irgb6 (wild-type, W3A mutant, and G277D/G285T/G286F mutant) docking with phospholipid polar head groups. See Fig S4 for detail. **(D)** Surface presentation of the PI5P pocket colored by elements. Blue: nitrogen, red: oxygen, and gray: carbon. The left side of the pocket where the PI5P head docks is covered with the hydrophilic/ionic residues, whereas the right side is covered with the hydrophobic residues (Trp3, Leu38, and Val42).

loop (Figs 4C and S4C and D). For the latter mutations, Gly277, Gly 285, and Gly 286 were substituted with aspartic acid, threonine, and phenylalanine, respectively, determined by reference to corresponding Irga6 residues (G277D/G285T/G286F). Consequently, the docking scores to W3A became worse for every phospholipid and the preference for PI5P was lost (Figs 4C and S4C). For the G277D/G285T/G286F mutant, PI5P was still preferred over other phospholipids, albeit the mean Glide Scores for the PI5P-mutant were worse than for PI5P-Irgb6 (Figs 4C and S4D). These docking results further support the specificity of PI5P to the Irgb6 pocket.

We also checked docking of the heads of four kinds of phospholipids with several lengths of glycerol backbone or acyl chain to Irgb6. Consequently, Glide Scores of PI5P were always lower

than those of PS, PE, or PC, suggesting PI5P is the best-suited phospholipid to be targeted by Irgb6 (Fig S4E and F). It should be noted that the short glycerol backbone or short acyl chains tended to direct themselves towards the hydrophobic pocket near Trp3. Considering that the lipid tails should be embedded in the bilayer membrane (Muftuoglu et al, 2016) some conformational change at the N-terminal helices should occur before rigid binding of Irgb6 to the PV membrane.

### In vivo evaluation of structural model for PVM binding

Docking simulations indicated that the head group of PI5P is on the PVM-binding loop and the central helix pair, thus the acyl chain should run towards the N-terminal helix αA. To assess the role of the putative membrane-binding region in Irgb6, we generated three Irgb6 mutants. The Irgb6 mutant in which all residues in the PVM-binding loop (277–286 amino acids) were entirely substituted with those of Irga6 was denoted Irgb6_a6(all). Same as the Glide docking experiments, Irgb6(G277D/G285T/G286F) and Irgb6(W3A) were also generated.

To test whether the Irgb6_a6(all), the G277D/G285T/G286F, or the W3A mutant was localized on *T. gondii* PVM, the Irgb6 mutants in addition to wild-type Irgb6 constructs were retrovirally overexpressed in Irgb6-deficient MEFs (Fig 5A). We confirmed that wild-type and mutant Irgb6 proteins were expressed at comparable levels in the reconstituted cells (Fig 5A), and that localization of Irgb6_mutants were similar to that of wild-type Irgb6 in uninfected cells (Fig S5A). Then we tested them for IFN-γ–induced reduction of *T. gondii* numbers and the recruitment to *T. gondii* PVM (Fig 5B and D). When IFN-γ–induced reduction of parasite numbers was examined, Irgb6-deficient MEFs reconstituted with wild-type Irgb6 were able to recover the IFN-γ–induced reduction of *T. gondii* numbers (Fig 5B). Reconstitution of wild-type Irgb6 in Irgb6-KO MEFs resulted in almost 20% *T. gondii* survival, which was largely similar to the parasite survival in IFN-γ–stimulated wild-type MEFs (Pradipta et al, 2021). In sharp contrast, Irgb6-KO MEFs that expressed the Irgb6_a6(all), the G277D/G285T/G286F, or the W3A mutants were not able to restore the IFN-γ–induced reduction of *T. gondii* numbers (Fig 5B). Furthermore, reconstitution of wild-type Irgb6 in Irgb6-deficient MEFs recovered the recruitment to *T. gondii* PVM, whereas that of the Irgb6_a6(all), the G277D/G285T/G286F, or the W3A mutants did not reconstitute the mutant recruitment (Figs 5C and D and S5B). Consistent with the previous finding that Irgb6 regulates loading of Irga6 and Irgb10 (Lee et al, 2020), reconstitution of wild-type Irgb6 but not of the Irgb6_a6(all), the G277D/G285T/G286F, or the W3A mutants increased percentages of Irga6-or Irgb10-localized vacuoles (Fig 5E–H). Collectively, a cluster of glycine residues in the PVM-binding loop and the N-terminal tryptophan are essential for the Irgb6 PVM targeting and the IFN-γ–induced cell-autonomous responses to *T. gondii*.

## Discussion

Irgb6 has a crucial role to target the PVM of *T. gondii* to facilitate its destruction. Our atomic structures of Irgb6 solved here elucidated the structural mechanisms of PVM recognition by Irgb6. Irga6 or

Irgb10 was reported to have the myristoylation site at their N-terminus (Haldar et al, 2013). Irgb6 uses different mechanisms in which Irgb6 binds to PI5P or PS to assess the PVM. The PI5P-binding site is located at the bottom surface that is composed of both the N- and C-domains, opposite to the GTP-binding pocket (Fig 6A).

Irgb6 maintains structural features common among IRGs. It is composed of three domains: N, G, and C; and, the nucleotide-dependent conformational change is also conserved in comparison to Irga6 structures. Our structures solved here do not represent the active GTP form in the pre-hydrolysis state. However, the purified Irgb6 protein used for the crystallization had GTPase activity (Fig S1B), suggesting that Irgb6 without any accessory proteins or cofactors can hydrolyze GTP. Thus, by analogy with Irga6 (Pawlowski et al, 2011), homodimerization of Irgb6 using the G-domain as an interface will activate the Irgb6 GTPase.

The nucleotide-dependent conformational change of G-domain transduces N- and C-domains through the linker helix αE (Fig 6B). This helix is located at the center among N-, G-, and C-domains and relays the conformational change of G2/switch I and G3/switch II to the N- and C-domains to generate the rotational movement known as a "power stroke" (Chappie et al, 2011). Our structures solved here represent the large conformational change of G4-G5 during GTP binding, making the top interface ready for homodimerization (Fig 6B). They also show small rotation of N- and C-domains via the helix αE movement that is induced by the small change of G2/switch I (Fig 6B). By the analogy with Irga6, homodimerization through the G-domains would change and stabilize the conformation of switch I and switch II. This would induce a power stroke of N- and C-domains relayed through helix αE, thus re-modeling the PV membrane. Further structural studies are needed to prove this hypothesis.

In contrast to the highly conserved GTPase domain, the helix αA in the N-domain and whole C-domain, both of which serve as a membrane-binding interface through PI5P binding, have large conformational variations among IRGs. In comparison with Irga6, the N- and C-domains of Irgb6 rotate in opposite directions, resulting in closure of the cleft between N- and C-domains (Fig 3B and C). The class specific tryptophan 3 residue at the N-terminus plays a pivotal role for the cleft closure between N- and C-domains, mainly made by hydrophobic interactions. Because of this movement, Irgb6 can produce an Irgb6-specific membrane-binding interface to recognize the PVM.

As described above, if we used the head domain of PI5P with the short acyl chain for the docking simulation, the acyl chains tended to direct themselves towards the hydrophobic pocket between the N- and C-domains. The acyl chain of the membrane-bound PI5P, on the other hand, must be oriented toward the PVM. To understand the conformational change during GTP binding through homodimerization, we compared the membrane-binding surface of Irgb6 with that Irga6 in the active GTP form in which conformational rearrangements of N- and C-domains were exclusively observed in Irga6. Interestingly, Irga6 had a wider open pocket than that of Irgb6 precisely at the putative acyl chain binding side of PI5P (Fig S6A). In this conformation, the acyl chain could interact with Irga6 through hydrophobic attractions, pass through the pocket, and then enter the membrane. We further checked the hydrophobic pocket of Irgb6. GTP-bound Irgb6 takes an alternative conformation at Trp3.

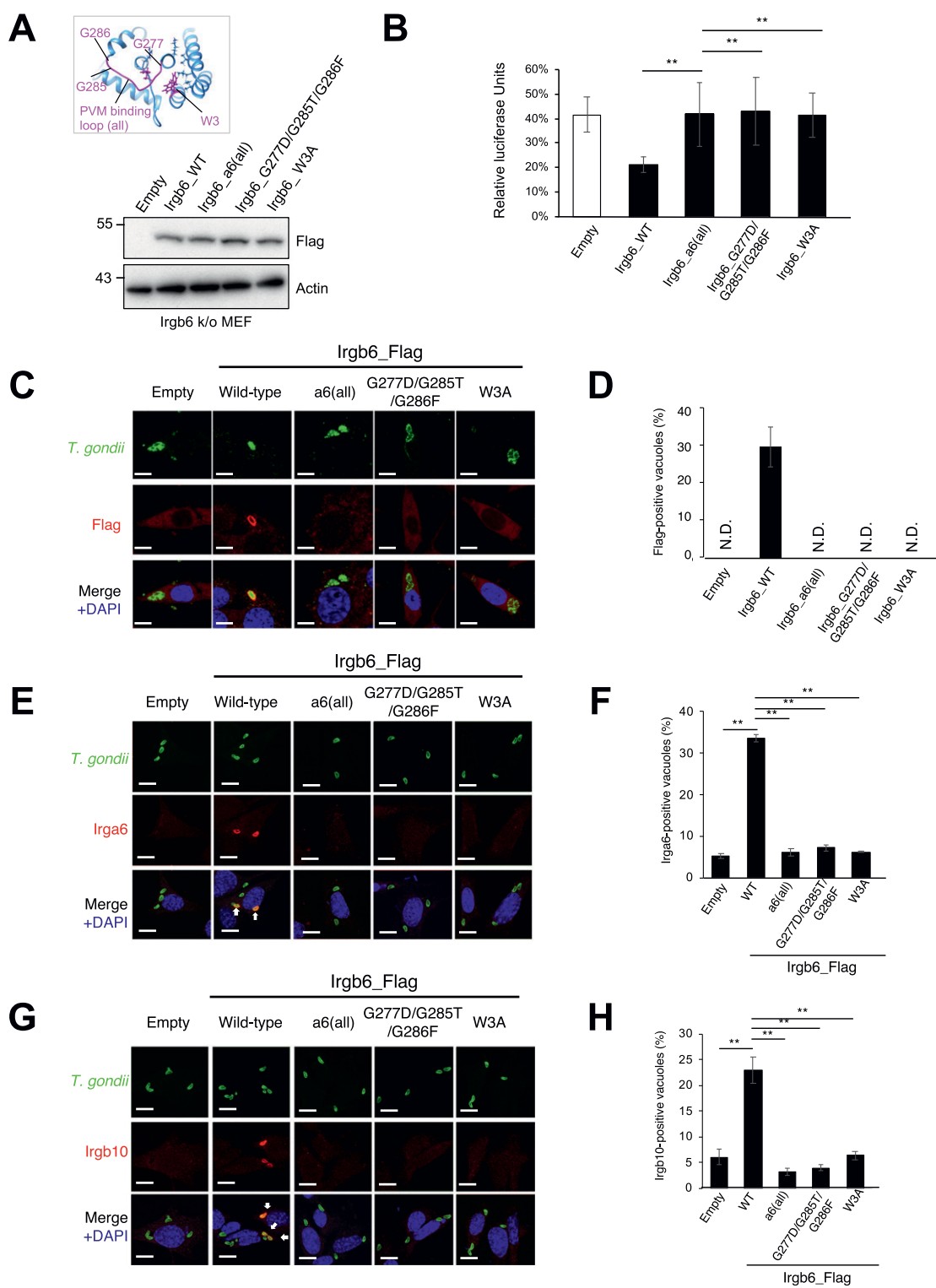

**Figure 5. The membrane-binding region is essential for Irgb6 accumulation on *Toxoplasma gondii* parasitophorous vacuole membrane.**
**(A)** Western blot image to detect stably expressed Irgb6 protein after retroviral transfection and puromycin selection. The mutation positions are indicated in the top panel. **(B)** *T. gondii* survival rate in the indicated Irgb6 reconstitution in Irgb6-KO MEFs with IFN-γ stimulation relative to those without IFN-γ treatment by luciferase analysis at 24 h post infection. All graphs show the mean ± SEM in three independent experiments. All images are representative of three independent experiments. N.D., not detected; **$P < 0.01$. *T. gondii* survival and Irgb6_Flag recruitment comparison between genotypes applied one-way ANOVA (Tukey's multiple comparisons test). White arrows to indicate recruitment of effector on *T. gondii* PV. Scale bars on microscope images represent 10 μm. **(C)** Confocal microscope images to show the localization of Irgb6-Flag (red) to *T. gondii* PV (green), and DAPI (blue) at 4 h post infection in IFN-γ–treated Irgb6-KO MEFs reconstituted with indicated Irgb6.

In these two alternative forms, the depths or sizes of the pocket openings differ significantly (Fig S6B and C). Thus, the conformation of the N-terminal helix could alter the shape and depth of hydrophobic pocket of Irgb6. From these observations, we assume that the GTPase activation by homodimerization of Irgb6 could change the helical domain to open the hydrophobic pocket to accommodate the acyl chain. Further structural studies are required to prove this hypothesis.

To clarify the specificity of PI5P to Irgb6, we additionally examined the glide docking of phospholipids to the Irga6 (PDB ID: 1TQ2) and Irgb10 (PDB ID: 7C3K). Surprisingly, both Irga6 and Irgb10 have potential to bind to phospholipids (Fig S4G–K). In particular, Glide scores of Irgb10 represented the lowest binding free energy among three IRGs, suggesting its high preference to the phospholipids. Nevertheless, the specificity for PI5P is unique in Irgb6. A preference to PS was observed in Irgb10, whereas Irga6 prefers PS or PE to PI5P. Therefore, the distributions of IRGs on PVM could be controlled by the preferences of phospholipids.

In the present study, we solved the atomic structure of Irgb6 monomer and elucidated the structural details of PVM-binding interface of Irgb6. Considering that Gbp1 regulates the localization or activity of Irgb6 on the PVM, biochemical and structural analyses of Gbp1-Irgb6 interaction are required to solve the molecular mechanisms of PVM disruption and pathogen clearance (Khaminets et al, 2010). Also, the rhoptry protein 18 (ROP18), a serine threonine kinase secreted by *T. gondii*, phosphorylates threonine residues in switch I of Irgb6 to disarm the innate clearance by host cells (Fentress et al, 2010; Steinfeldt et al, 2010). By elucidating the structural mechanisms of how ROP18 inactivates Irgb6, therefore, the whole picture of host cell-autonomous immunity and microbial counter-defense system will be unveiled.

# Materials and Methods

### Protein expression

The full-length *Mus musculus* Irgb6 gene (*Tptg 2*, Gene ID: 100039796) was PCR-amplified with specific primers (5′-<u>GAAGTTCTGTTCCAGGGGCC</u>-<u>C</u>ATGGCTTGGGCCTCCAGC-3′ and 5′-<u>CGATGCGGCCGCTCGAG</u>TTAT-CAAGCTTCCCAGTACTCGG-3′; the original sequence of pGEX-6P-1 are underlined) from the pWT_Irgb6_full (Lee et al, 2020) and then subcloned into the directly downstream of PreScission protease site of pGEX-6P-1 (Cytiva) by Gibson Assembly system (New England Biolabs Inc.) to create pRN108. The pRN108 was transformed into *Escherichia coli* strain BL21(DE3). The transformant were grown in LB medium with 50 mg/l ampicillin at 25°C to an OD600 nm of 0.4, and GST-tagged Irgb6 was expressed overnight with final 0.1 mM isopropyl β-D-1-thiogalactopyranoside. The cells were harvested and stored at –80°C.

### Protein purification

Irgb6 was purified at 4°C. The frozen BL21(DE3) cells were suspended in solution-I (50 mM HEPES-KOH, pH 7.5, 150 mM NaCl, 2 mM dithiothreitol, 0.7 $\mu$M leupeptin, 2 $\mu$M pepstatin A, 1 mM phenyl-methylsulfonyl fluoride, and 2 mM benzamidine) and sonicated on ice. The cell lysate was centrifuged (80,000$g$, 30 min) and GST-Irgb6 in the soluble fraction was purified by affinity chromatography using a Glutathione Sepharose 4B column (Cytiva) equilibrated with solution-I. The GST domain of the protein was cleaved by overnight incubation with GST-tagged HRV 3C protease (homemade) on the resin. The free Irgb6 which contains two extra N-terminal residues, Gly–Pro, was eluted with solution-I and was concentrated to with an Amicon Ultra 10-kD MWCO concentrator (Merck Millipore). The protein was further subjected to SEC on a HiLoad 16/600 Superdex 75 column pg column (Cytiva) equilibrated in solution-II (50 mM HEPES-KOH, pH 7.5, 150 mM NaCl, 5 mM MgCl$_2$, and 2 mM dithiothreitol). Peak fraction containing Irgb6 at ~47 kD elution position was concentrated using the concentrator for crystallization. Protein concentration was estimated by assuming an A280 nm of 0.916 for a 1 mg/ml solution.

### Crystallization

Nucleotide-free Irgb6 crystals diffracting to 1.9 Å resolution were obtained from sitting drops with a 12 mg/ml protein solution and a reservoir solution consisting of 0.1 M MIB buffer, pH 6.0 (Molecular Dimensions), 25% Polyethylene Glycol 1500 (Molecular Dimensions) at 20°C. GTP-binding Irgb6 crystals diffracting to 1.5 Å resolution were obtained from sitting drops with a 9 mg/ml protein solution containing 2 mM GTP (Roche) and a reservoir solution consisting of 0.1 M Sodium Citrate buffer, pH 5.4 (Wako), and 18% (wt/vol) polyethylene glycol 3350 (Sigma-Aldrich) at 20°C.

### Data collection and structure determination

Single crystals were mounted in LithoLoops (Protein Wave) with the mother liquor containing 10% (vol/vol) or 20% (vol/vol) glycerol as a cryoprotectant and were frozen directly in liquid nitrogen before X-ray experiments. Diffraction data collection was performed on the BL32XU beamline at SPring-8 using the automatic data collection system ZOO (Hirata et al, 2019). The diffraction data were processed and scaled using the automatic data processing pipeline KAMO (Yamashita et al, 2018). The structure was determined using PHENIX software suite (Liebschner et al, 2019). Initial phase was solved by molecular replacement using PDB ID: 1TQD, 1TQ2, and 1TPZ with phenix.phaser. The initial model was automatically constructed with phenix.AutoBuild. The model was manually built with Coot (Emsley & Cowtan, 2004) and refined with phenix.refine and Refmac (Vagin et al, 2004) in CCP4 software suite (Winn et al, 2011). The statistics of the data collection and the structure refinement are summarized in Table S1. UCSF Chimera (Pettersen et al, 2004) was used to create images,

---

**(D)** Recruitment percentages of Irgb6_Flag. **(E)** Confocal microscope images to show the localization of Irga6 (red) to *T. gondii* PV (anti-GRA2; green), and DAPI (blue) at 4 h post infection in IFN-γ–treated Irgb6-KO MEFs reconstituted with indicated Irgb6. **(F)** Recruitment percentages of Irga6. **(G)** Confocal microscope images to show the localization of Irgb10 (red) to *T. gondii* PV (anti-GRA2; green), and DAPI (blue) at 4 h post infection in IFN-γ–treated Irgb6-KO MEFs reconstituted with indicated Irgb6. **(H)** Recruitment percentages of Irgb10. All graphs show the mean ± SEM in three independent experiments. All images are representative of three independent experiments. Recruitment percentages of indicated effectors calculated by counting almost 100 *T. gondii* PV in one experiment were shown as results of three independent experiments. N.D., not detected; **$P$ < 0.01. *T. gondii* survival and Irgb6_Flag recruitment comparison between genotypes applied one-way ANOVA (Tukey's multiple comparisons test). White arrows to indicate recruitment of effector on *T. gondii* PV. Scale bars on microscope images represent 10 $\mu$m.

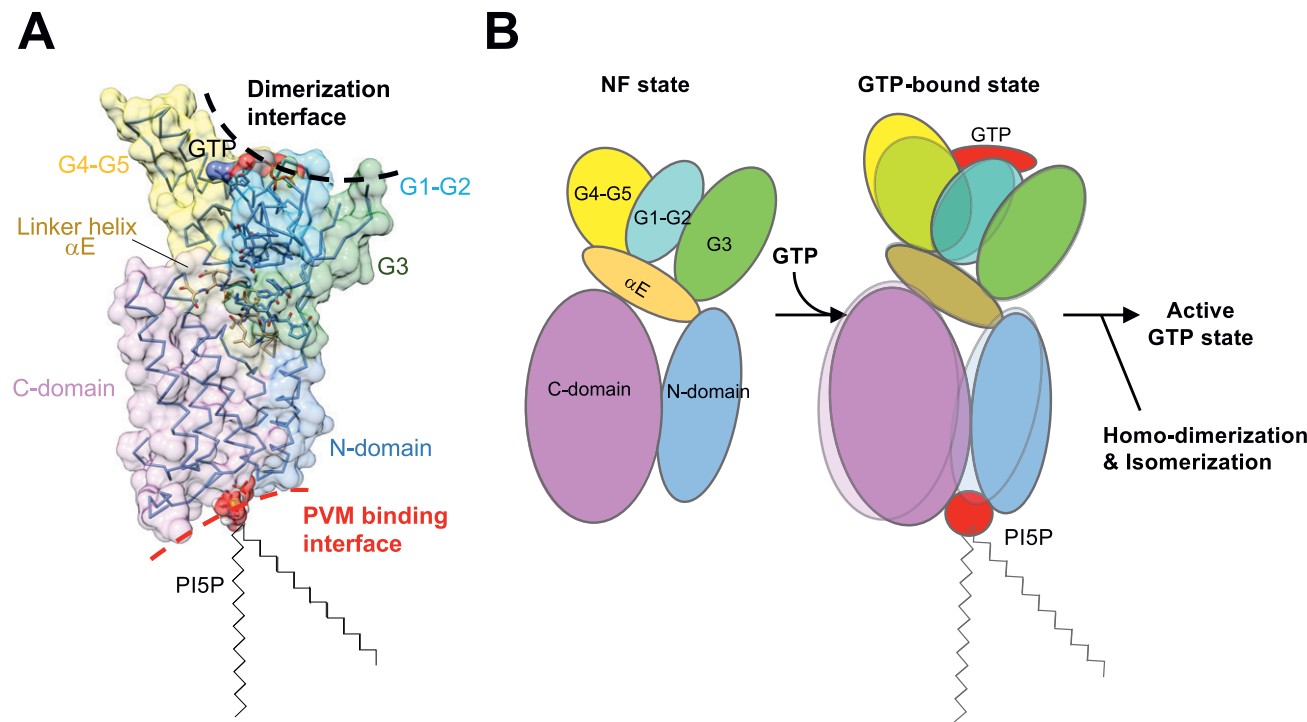

**Figure 6. Structural model of parasitophorous vacuole membrane recognition during the GTP binding.**
**(A)** Structure of Irgb6 in the GTP-bound state. **(A, B)** Schematic model of conformational change of Irgb6 during GTP binding, shown with the same colors in the panel (A).

compare structures, and calculate RMSDs. RMSDs were calculated using MatchMaker in UFSC Chimera that were based on the structure-based multiple sequence alignment. RMSDs of whole structures were calculated by aligning all Cα atom pairs from two proteins. RMSDs of each N-, G-, or C-domains were also calculated by aligning Cα atom pairs in N-, G-, or C-domains from two proteins.

## Analysis of nucleotide component

Irgb6 and GTP were prepared 40 mM in 50 mM HEPES-KOH, pH 7.5, 1 mM MgCl$_2$, 1 mM EGTA-KOH, pH 7.0, and 150 mM NaCl. A 25 µl Irgb6 sample were mixed to equal volume of GTP sample and incubated at 36°C for 30 min. A 1 ml of 8 M urea was added to the mixture and heated at 95°C for 1 min, followed by ultrafiltration using Amicon Ultra-0.5 10-kD MWCO concentrator (Merck Millipore). A 900 µl of the solution that passed through the ultrafiltration membrane was analyzed by anion exchange chromatography using a Mono Q 5/50 Gl column (Cytiva) equilibrated with 50 mM HEPES-KOH, pH 7.0. Components of the reaction mixture, GTP and GDP, were completely separated by elution with 0–0.2 M NaCl gradient in 50 mM HEPES-KOH, pH 7.0. Fresh GTP (Nacalai Tesque) and GDP (WAKO) were used to confirm the elution position. A control experiment was performed using the reaction buffer.

## In silico docking simulation

Molecular docking was performed using Schrödinger suite. The 2D structures of the four phospholipid ligands, PI5P, PS, PE, and PC were obtained from PubChem (https://pubchem.ncbi.nlm.nih.gov/) (Fig S4). Acyl chains were truncated up to their corresponding polar head groups. Ligands were also prepared with the polar heads and glycerol backbones, as well as with 4 and 16 carbon acyl chains. The free ligands were converted to three-dimensional structures and their geometries were optimized with the correct chirality using Ligprep. LigPrep was also used to produce different conformations for each ligand structure. Before docking, the Irgb6 protein was prepared using the protein preparation wizard. Subsequently, a grid box was centered on acids Trp3, Lys275, and Arg371. Three similar grid centers and two positions of Trp3 and Arg371 were independently considered (Fig S5). The prepared ligands were docked with the preprocessed Irgb6 protein grids using Glide standard precision (SP) docking mode with flexible ligand sampling.

## Cells and parasites

MEFs that lack Irgb6 are described previously (Lee et al, 2020). Irgb6-deficient MEFs were maintained in DMEM (Nacalai Tesque) supplemented with 10% heat-inactivated FBS (Gibco, Life Technologies), 100 U/ml penicillin (Nacalai Tesque), and 100 µg/ml streptomycin (Nacalai Tesque). The complete medium comprised 10% heat-inactivated FBS in RPMI 1640 medium (Nacalai Tesque). *T. gondii* were parental PruΔHX, luciferase-expressing PruΔHX. They were maintained in Vero cells by passaging every 3 d in RPMI 1640 supplemented with 2% heat-inactivated FBS, 100 U/ml penicillin, and 100 µg/ml streptomycin.

## Reagents

Antibodies against FLAG M2 (F3165), and β-actin (A1978) were obtained from Sigma-Aldrich. Anti-Irga6 (10D7) and -Irgb10 rabbit

polyclonal antibodies were kind gifts of Dr. Jonathan C Howard. Anti-GRA7 rabbit polyclonal or anti-GRA2 mouse monoclonal antibodies to staining *T. gondii* PV were kind gifts from Drs. John Boothroyd or Dominque Soldati-Favre, respectively. Anti-KDEL (1D5) was obtained from MBL.

### Cloning and recombinant expression

The region of interest of the cDNA corresponding to the wild-type, indicated point mutants or deletion mutants of Irgb6 (GenBank accession no. NM_001145164) were synthesized from the mRNA of the spleen of C57BL6 mice using primers Irgb6_F 5′-gaattcaccATGGCTTGGGCCTC CAGCTTTGATGCATTCT-3′ and Irgb6_R 5′-gcggccgcTCActcga gAGCTTCCCAGTACTCGGGGGGGCTCAGATAT-3′. Irgb6_a6(all), G277D/G285T/G286F, and W3A mutants were generated using primers a6(all)_F 5′-TCTTCCTAGAAGCCATGAAGGCTgacctagtgaatatcatcccttctctgacctttATG ATCAGTGATATCTTAGAGAAT-3′ and a6(all)_R 5′-ATTCTCTAAGA TATCACTGATCATAaaggtcagagaagggatgatattcactaggtcAGCCTTCATG GCTTCTAGGAAGA-3′; G277D/G285T/G286F _F 5′-GTCTTCCTAGA AGCCATGAAGGCTGacGCATTAGCCACCATTCCACTTaactttATGATCAGT GATATCTTAGAGAATCT-3′ and G277D/G285T/G286F _R 5′-AG ATTCTCTAAGATATCACTGATCATAaaagttAAGTGGAATGGTGGCTAATGCgt CAGCCTTCATGGCTTCTAGGAAGAC-3′; W3A _F 5′-gaattcaccATGGC TgcGGCCTCCAGCTTTGATGCATTCTTTAAGAATTT-3′ products were ligated into the EcoRI/XhoI site of the retroviral pMRX-Flag expression vector for retroviral infection. The sequences of all constructs were confirmed by DNA sequencing.

### Western blotting

MEFs were stimulated with IFN-γ (10 ng/ml) overnight. The cells were washed with PBS and then lysed with 1× TNE buffer (20 mM Tris–HCl, 150 mM NaCl, 1 mM EDTA, and 1% NP-40) or Onyx buffer (20 mM Tris–HCl, 135 mM NaCl, 1% Triton-X, and 10% glycerol) for immunoprecipitation, which contained a protease inhibitor cocktail (Nacalai Tesque) and sonicated for 30 s. The supernatant was collected, incubated with the relevant antibodies overnight, and then pulled gown with Protein G Sepharose (GE) for immunoprecipitation. Samples and/or total protein was loaded and separated in 10% or 15% SDS–PAGE gels. After the appropriate length was reached, the proteins in the gel were transferred to a polyvinyl difluoride membrane. The membranes were blocked with 5% dry skim milk (BD Difco Skim milk) in PBS/Tween 20 (0.2%) at room temperature. The membranes were probed overnight at 4°C with the indicated primary antibodies. After washing with PBS/Tween, the membranes were probed with HRP-conjugated secondary antibodies for 1 h at room temperature and then visualized by Luminata Forte Western HRP substrate (Millipore).

### Measurement of *T. gondii* numbers by a luciferase assay

The number of luciferase-expressing *T. gondii* was indirectly counted by the luciferase units (Yamamoto et al, 2012). Cells were untreated or treated with IFN-γ (10 ng/ml) for 24 h. After the stimulation, the cells were infected with luciferase-expressing PruΔHX *T. gondii* (MOI of 0.5) for 24 h. The infected cells were collected and lysed with 100 μl of 1× passive lysis buffer (Promega).

The samples were sonicated for 30 s before centrifugation and 5 μl of the supernatants were collected for luciferase expression reading by the dual-luciferase reporter assay system (Promega) using a GLOMAX 20/20 luminometer (Promega). The in vitro data are presented as the percentage of *T. gondii* survival in IFN-γ–stimulated cells relative to unstimulated cells (control).

### Immunofluorescence microscopy

MEFs were uninfected or infected with *T. gondii* (MOI 5 or 2) after stimulation with IFN-γ (10 ng/ml) for 24 h. The cells were infected for the indicated time in the respective figures and then fixed for 10 min in PBS containing 3.7% formaldehyde. Cells were then permeabilized with PBS containing 0.002% digitonin (Nacalai Tesque) and blocked with 8% FBS in PBS for 1 h at room temperature. Next, the cells were incubated with antibodies relevant to the experiments for 1 h at 37°C. After gently washing the samples in PBS, the samples were incubated with Alexa 488– and 594–conjugated secondary antibodies as well as DAPI for 1 h at 37°C in the dark. The samples were then mounted onto glass slides with PermaFluor (Thermo Fischer Scientific) and observed under a confocal laser microscope (FV1200 IX-83; Olympus). Images are shown at ×1,000 magnification (scale bar 10 μm). To measure recruitment rates, 100 vacuoles were observed and the numbers of vacuoles coated with effectors were calculated. The counting was repeated three times (three technical replicates). The mean of the three technical replicates was calculated and shown in each circle. The independent experiments were repeated three times (three biological replicates).

### Statistical analysis

Three points in all graphs represent three means derived from three independent experiments (three biological replicates). All statistical analyses were performed using Prism 9 (GraphPad). In assays for *T. gondii* survival and recruitment of Irgb6_Flag, Irga6, or Irga6, ordinary one-way ANOVA was used when there were more than two groups.

## Data Availability

The crystal structure data of Nucleotide-free Irgb6 and GTP-bound Irgb6 in this study have been registered in the Protein Data Bank (PDB) on PDBID 7VES and PDBID 7VEX, respectively.

### Online supplemental material

Fig S1A and B shows that Irgb6 expressed in *E. coli* was purified as a monomer and has GTPase activity. Fig S2A and B shows the GTP pockets of Irgb6_GTP. Fig S2C shows the GTP pockets of Irgb6_NF. Fig S3A shows amino acid sequence alignment among Irgb6, Irga6, and Irgb10. Fig S3B shows the structural homology between Irgb6_NF and Irga6_NF. Fig S3C shows the crystal packing of Irgb6_NF. Fig S4A shows structures of PI5P, PS, PE, and PC. Fig S4B–S4K shows the detailed Glide scores by in situ docking simulations. Fig S5A shows localization of Irgb6 wild-type and mutants in uninfected cells. Fig S5B shows raw data for Fig 5C. Fig S6A shows PVM-binding site of

Irga6 in the active GTP-form represents widely open pocket. Fig S6B and C shows PVM-binding pocket of Irgb6 in the GTP-bound form represents semi closed form and closed form, respectively.

## Supplementary Information

## Acknowledgements

We thank K Chin for assistance and other colleagues for discussions. We also thank Drs. JC Howard, JC Boothroyd, and D Favre-Soldati for antibodies. This work was supported by the Japan Agency for Medical Research and Development (AMED) (JP20fk0108137 [M Yamamoto], JP20wm0325010 [M Yamamoto], JP20jm0210067 [M Yamamoto], JP20am0101108 [DM Standley], and JP20gm0810013 [R Nitta]). We acknowledge support from the Japan Society for the Promotion of Science (21K06988 [Y Saijo-Hamano], 20B304 [M Yamamoto], 19H04809 [M Yamamoto], 19H00970 [M Yamamoto], 19H03396 [R Nitta], 21H05254 [R Nitta], and 21K19352 [R Nitta]) and from JST (Moonshot R&D) (Grant Number JPMJMS2024). This research was also supported by Platform Project for Supporting Drug Discovery and Life Science Research (Basis for Supporting Innovative Drug Discovery and Life Science Research [BINDS]) from AMED under Grant Number JP19am0101070 (support number 2057). We also acknowledge support from the Hyogo Science and Technology Association to R Nitta.

### Author Contributions

Y Saijo-Hamano: conceptualization, data curation, formal analysis, funding acquisition, validation, investigation, visualization, and writing—original draft, review, and editing.
AA Sherif: data curation, formal analysis, validation, investigation, and visualization.
A Pradipta: data curation, formal analysis, and investigation.
M Sasai: data curation, formal analysis, and investigation.
N Sakai: data curation, formal analysis, supervision, validation, investigation, and visualization.
Y Sakihama: data curation and investigation.
M Yamamoto: conceptualization, data curation, formal analysis, supervision, funding acquisition, validation, investigation, visualization, project administration, and writing—original draft, review, and editing.
DM Standley: formal analysis, supervision, investigation, project administration, and writing—original draft, review, and editing.
R Nitta: conceptualization, formal analysis, supervision, funding acquisition, visualization, project administration, and writing—original draft, review, and editing.

### Conflict of Interest Statement

The authors declare that they have no conflict of interest.

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
