## [Reviewer comments · Life Science Alliance]

Life Science Alliance

Structural basis of membrane recognition of *Toxoplasma gondii* vacuole by Irgb6

Yumiko Saijo-Hamano, Aalaa Sherif, Ariel Pradipta, Miwa Sasai, Naoki Sakai, Yoshiaki Sakihama, Masahiro Yamamoto, Daron Standley, and Ryo Nitta

DOI: <https://doi.org/10.26508/lsa.202101149>

Corresponding author(s): Ryo Nitta, Kobe University Graduate School of Medicine

Review Timeline:

Submission Date:	2021-07-06
Editorial Decision:	2021-07-29
Revision Received:	2021-09-16
Editorial Decision:	2021-10-05
Revision Received:	2021-10-08
Editorial Decision:	2021-10-11
Revision Received:	2021-10-15
Accepted:	2021-10-18

Scientific Editor: Novella Guidi

Transaction Report:

July 29, 2021

Re: Life Science Alliance manuscript #LSA-2021-01149-T

Prof. Ryo Nitta
Kobe University Graduate School of Medicine
Division of Structural Medicine and Anatomy
7-5-1 Kusunoki-cho
Chuo-ku
Kobe, Hyogo 650-0017
Japan

Dear Dr. Nitta,

Thank you for submitting your manuscript entitled "Structural basis of membrane recognition of *Toxoplasma gondii* vacuole by Irgb6" to Life Science Alliance. The manuscript was assessed by expert reviewers, whose comments are appended to this letter. As you will note from the reviewers' comments below, all the reviewer are quite positive and excited about the work that in their view constitutes a significant advance in the field of *Toxoplasma* infection defense. However, they do raise some concerns that would need to be addressed in the revised version before resubmission. We, thus, encourage you to submit a revised version of the manuscript back to LSA that responds to all of the reviewers' points including conducting a loop-swap experiment in cells with the equivalent loop in Irgb10 to prove the unique nature of this phospholipid binding by Irgb6. Alternatively, provide Glide modelling as suggested by Reviewer 1. Please also provide the GMPPNP-bound Irgb6 structure that is mentioned in one suppl figure and in the methods, but not in the main text, as suggested by reviewer 1. We also encourage you, in line with reviewer 2, to provide the occupancy, the B factor for the ligand, as well as the Molprobit analysis for the ligand and the adjacent Mg²⁺, to prove the GTP ligand. As suggested by reviewer 3, please also provide a co-stain for ER markers to demonstrate that these amino acids are essential for PV localization.

Thank you for this interesting contribution to Life Science Alliance. We are looking forward to receiving your revised manuscript.

Sincerely,

B. MANUSCRIPT ORGANIZATION AND FORMATTING:

Reviewer #1 (Comments to the Authors (Required)):

This manuscript presents high resolution X-ray crystallography structures of the host defence protein Irgb6 in different nucleotide states (Irgb6-GTP and Irgb6-NF). This protein is one of the first Irg proteins recruited to the parasite *Toxoplasma gondii* parasitophorous vacuole membrane (PVM) during host defence against the parasite. The authors compare the structures with previous structures of related Irga6 and Irgb10. This family of proteins consists of three domains, the N terminal domain, the G domain and the C terminal domain. The N terminal domain is similar in all three Irg proteins. The GTPase G domain of Irgb6-GTP is surprisingly more similar to the previous Irga6-GDP structure rather than the Irga6-GMPPNP structure. The C terminal domain of Irgb6 is the most divergent compared to the other Irg structures, with a distinct rotation of outer helices in the C terminal domain for Irgb6. The authors then run simulations to see how lipids bind to Irgb6, showing that the phospholipid PI5P binds to an interface across the N and C terminal domain. This interface is then validated in cells by mutating individual residues, as well as replacing the Irgb6 loop with the equivalent loop from Irga6. These mutations all fail to bind to the *Toxoplasma* membrane and rescue *Toxoplasma*-infected Irgb6-deficient MEF cells. This suggests that they have successfully identified important lipid-binding residues that are important to the host defence response against *Toxoplasma* infection.

In general this manuscript is well written, containing a good set of experiments providing a structural basis for phospholipid recruitment of Irgb6, and the importance of this in defence against *Toxoplasma*. The conclusions that the authors make are well supported by the data, and their findings constitute a significant advance in the field of *Toxoplasma* infection defence. However there are two important issues that need to be addressed, detailed below. Also there are a couple of minor issues that should be addressed to better contextualise the Irgb6 phospholipid binding and its role in *Toxoplasma* infection defence.

Major points:

1. In Figure S1C and in the methods, there is a third structure mentioned, a structure of GMPPNP-bound Irgb6, including data collection and model refinement parameters. However this is not mentioned elsewhere in the text. Do the authors have this structure? If they do, could the authors include this structure in the manuscript and compare it to the other Irg structures, particularly the previous Irg6a-GMPPNP structure?
2. The modelling results combined with the loop-swap experiment give good evidence that Irgb6-PVM-loop equivalent in Irga6 cannot mediate phospholipid binding. I am interested as to whether the same is true of Irgb10. Could the authors conduct a similar loop-swap experiment in cells with the equivalent loop in Irgb10? I realise this may take a month or two if this mutant has not already been generated. As an alternative, could the authors use Glide modelling to see if the phospholipid could bind to Irgb10 in the same place? Presuming Irgb10 also cannot mediate phospholipid binding, this would drive home the unique nature of this phospholipid binding by Irgb6.

Minor points:

1. The authors performed the lipid-binding modelling on the GTP-bound Irgb6 structure. Could the authors clarify in the text whether they expect the lipid binding to occur in the nucleotide-free state of Irgb6? If they do not expect lipids to bind in this state, are there structural differences that support this?
2. Could the authors explicitly state in the text whether PI5P/PS binding is unique to Irgb6 or is expected in other Irg family members?
3. The authors state that the G-domain of Irgb6-GTP is more similar to Irg6a-GDP compared to Irg6a-GMPPNP, giving the

explanation that the Irg6b G2/G3 sequences do not coordinate the gamma phosphate, so overall the Irgb6-GTP G-domain recognizes the "GDP part" of the GTP. If one excludes the G2/G3 sequences of the Irg6a-GMPPNP structure, does it better align with the Irg6b-GTP structure, or is it still worse compared with the Irg6a-GDP structure?

4. We thank the authors for providing the crystal structures. These were, on the whole, well built and refined, particularly around the areas that the authors refer to in the manuscript. However there are a few regions where the model does not explain the density well, that should be addressed before release from the PDB. Irgb6-GTP: 145-148; 210-212; 359-362; 400-403. Irgb6-NF: 209-212.

5. Could the authors provide more detail on RMSD calculation in the methods? I am presuming, for example, they aligned the G domain of one structure to the G domain of another and reported the RMSD, rather than aligning the structure as a whole, then reporting the G domain RMSD, but this should be explicitly mentioned.

6. Could the authors clarify in the manuscript whether the Irgb6 rescue constructs in the cell experiments were overexpressed?

7. What is the killing activity of wild type IFN-gamma-stimulated MEFs for toxoplasma infection? Could the authors state this in the text for comparison?

8. Figure 4A&B: the lipid molecules are not very obvious in this figure. Could the authors make them stand out more by, for example, making the sticks in each lipid molecule thicker?

9. The authors state that the G domain in the GTP-bound structure is in a pre-isomerization state, referencing two papers describing an ATPase in a similar state. Are there any examples of other GTPases in this pre-isomerization state? Could authors include these?

A few parts of the manuscript require minor language clarification. Could the authors rephrase the following:

1. Page 2 Line 3: "it disrupts it". The usage of 'it' for two different objects is confusing.

2. Page 2 Line 8-9: "It is recruited.... and disrupts it". The usage of 'it' for two different objects is confusing.

3. Page 10 bottom paragraph: "Observed from the bottom side, the other side of the GTP-pocket...". The phrase "the other side of the GTP-pocket is confusing". Consider "the opposite face to the G domain binding face" or similar.

4. Page 10 bottom paragraph: "The C-domain helices rotate....". I'm presuming this is in the authors' structures, but it should be explicitly mentioned to avoid confusion.

5. Page 12 second paragraph: "The Inositol makes...". Lower case "i" in "inositol".

6. Page 15 Line 1: "Irgb6 has a crucial role to target the PVM...and destroy them". This implies that the Irgb6 destroys the PVM of *T. gondii*. If this is not the case could the authors rephrase this sentence?

Reviewer #2 (Comments to the Authors (Required)):

The manuscript by Yumiko Saijo-Hamano and colleagues report the crystal structures of irgb6 from mouse. Irga6, irgb10 and irgb6 control intracellular *Toxoplasma* growth in interferon activated murine cells. Here authors solved the last structure of these major three ISGs, modeled the structure complexed with phospholipids and validated the model by point mutations, thus partially explained the PVM binding mechanism. The work is original and interesting but needs some revisions and serious proof-reading before publication.

Major findings include:

Authors solved the crystal structures of the last IRG among three major IRGs (irga6, irgb6 and irgb10) which all can be recruited on the PVM.

Modeling and cellular assays suggested a different structural basis of membrane recognition by irgb6, apart from previously reported irga6 and irgb10.

Major issues

Authors claims a GTP-bound structure in Fig 1. However, according to the density map in Fig S2a, I am not convinced the ligand is GTP. The author need to provide the occupancy, the B factor for the ligand, as well as the Molprobit analysis for the ligand and the adjacent Mg²⁺. The twisted triphosphate tail seems unlikely as far as I can tell from Fig S2. Or is there a reasonable explanation about its orientation?

I am not sure that the PI5P head is shown when looking at Fig 4a, especially in Fig 4d where only C4 ring can be clearly seen.

Please make sure that the ligand is PI5P or adjust the panel for better visualization.

It would be nice to see simultaneous IFA staining for Irga6 or Irgb10 in Fig 5B. Do the mutations affect recruitments of other IRG proteins?

It is mentioned that Toxoplasma inactivates IRGs by phosphorylation of the G loop SWI. This statement should also include the work from the Howard lab (PMID: 21203588) Since the G loop is described as being different in the current model, it would be important to discuss these prior findings in light of the new structure. Does the altered conformation of the G loop make this region more or less accessible? Can the authors comment on how PO4 might alter the conformation to either prevent binding of GTP or hydrolysis?

Minor issues

In Fig1A, cyan was used to color N-domain but also for the general GTP-bound form. Cyan was also used to color G-domain in Fig1C, which was colored as green in Fig1A. I suggest recoloring N-domain in Fig1A to avoid confusing.

The completeness of the nucleotide-free data is low (especially when comparing with other two datasets) in the last shell. The reflection of a nominal high resolution here has a comparatively big difference between the R-work and R-free, especially in terms of the 1.8Å resolution. Authors might need to be more conservative in estimating the resolution to create a map with less noise and more accuracy.

In the last paragraph of the results (very inconvenient because of lacking line numbers), authors are referring Fig 5b that is describing irgb6 recruitment. Fig 5b and 5d need to be changed reciprocally according to authors description in the paper.

There is no Fig S4 while it was referred in the 'in situ docking stimulation' section.

No Fig S5 as well.

Ghosh, Molecular Cell, 2004, reference duplicated.

Reviewer #3 (Comments to the Authors (Required)):

This is a timely and well written structure-function report that informs the mechanism of IRGb6-membrane interaction. The study is important to our understanding of cell autonomous immune sensing for a range of vacuole-resident pathogens. In this way, it is appropriate for the readership of LSA. IRGb6 belongs to the family of p47 GTPases. Other members of this family, namely IRGa6 and Irgb10, have had their crystal structures solved. These proteins use a myristoylated glycine to interact with target membranes. However, this motif is not conserved in the paralog IRGb6. Instead, the C-terminal amphipathic domain has been shown to mediate phospholipid interactions, however, the structural mechanism of membrane binding remains is not known. This paper addresses this information gap by solving the first structures of IRGb6 in the GTP (1.5 Å) bound and unbound state (1.9 Å).

The central findings of the paper are that the G-domain, which contains the nucleotide binding pocket, undergoes a similar catalytic shift to other p47 IRGs based on structure comparison. This is consistent with the existing model where GTPase activity regulates homo-dimerization. In contrast to IRGb10 and IRGa6, the C-terminal domain of IRGb6, containing the alphaF-alphaGa "PVM binding loop," also undergoes a conformational shift in the GTP bound state. The GTP bound structure was used to model the interaction between the alphaF-alphaGa loops and phospholipids. PI5P was predicted to have the lowest binding free energy, suggesting this is a preferential interaction over PS, PE or PC. These results (Figure 4) somewhat recapitulate PIP strip data from (Lee et al 2019), however, in that study PE and PC binding were not observed, which could be better discussed.

To evaluate the significance of the PVM binding loop, a chimera between IRGb6 and IRGa6 was generated where the region between 277-286 was swapped, the 3 glycine residues were mutated, or the N-terminus tryptophan was substituted with alanine. The conclusion was that the glycine residues are necessary for optimal PVM localization and parasite restriction. However, additional controls seem important to show that the introduced point mutations in IRGb6 specifically inhibit phospholipid binding. For example, do these constructs lose PIP5 and PS binding specificity using PIP strips or a related approach (Lee et al 2020)? A related concern is that it is possible that loss of PVM localization/parasite restriction is simply due to misfolding of the mutant constructs in the ER. A co-stain for ER markers would help limit this concern and strengthen a model wherein these amino acids are essential for PVM localization.

Minor comments

-In Figure 5, a schematic, showing the position of point mutations along the protein would be useful to the reader.

-Including a statistic regarding the number of vacuoles and parasites evaluated would be useful to assess the robustness of the data presented in 5C-D.

Reviewer #1 (Comments to the Authors (Required)):

This manuscript presents high resolution X-ray crystallography structures of the host defence protein Irgb6 in different nucleotide states (Irgb6-GTP and Irgb6-NF). This protein is one of the first Irg proteins recruited to the parasite *Toxoplasma gondii* parasitophorous vacuole membrane (PVM) during host defence against the parasite. The authors compare the structures with previous structures of related Irga6 and Irgb10. This family of proteins consists of three domains, the N terminal domain, the G domain and the C terminal domain. The N terminal domain is similar in all three Irg proteins. The GTPase G domain of Irgb6-GTP is surprisingly more similar to the previous Irga6-GDP structure rather than the Irga6-GMPPNP structure. The C terminal domain of Irgb6 is the most divergent compared to the other Irg structures, with a distinct rotation of outer helices in the C terminal domain for Irgb6. The authors then run simulations to see how lipids bind to Irgb6, showing that the phospholipid PI5P binds to an interface across the N and C terminal domain. This interface is then validated in cells by mutating individual residues, as well as replacing the Irgb6 loop with the equivalent loop from Irga6. These mutations all fail to bind to the *Toxoplasma* membrane and rescue *Toxoplasma*-infected Irgb6-deficient MEF cells. This suggests that they have successfully identified important lipid-binding residues that are important to the host defence response against *Toxoplasma* infection.

In general this manuscript is well written, containing a good set of experiments providing a structural basis for phospholipid recruitment of Irgb6, and the importance of this in defence against *Toxoplasma*. The conclusions that the authors make are well supported by the data, and their findings constitute a significant advance in the field of *Toxoplasma* infection defence. However there are two important issues that need to be addressed, detailed below. Also there are a couple of minor issues that should be addressed to better contextualise the Irgb6 phospholipid binding and its role in *Toxoplasma* infection defence.

First of all, we thank this reviewer for deep understanding of our Irgb6 structure paper.

Major points:

1. In Figure S1C and in the methods, there is a third structure mentioned, a structure of GMPPNP-bound Irgb6, including data collection and model refinement parameters. However this is not mentioned elsewhere in the text. Do the authors have this structure? If they do, could the authors include this structure in the manuscript and compare it to the other Irg structures, particularly the previous Irga6-GMPPNP structure?

We have the structure of Irgb6 crystallized in the presence of GMPPNP. However, we have decided not to present it in this manuscript for the following reasons.

- 1) Main reason: This structure apparently takes the same conformation as GTP-bound Irgb6. Hence, greater structural or biological insights cannot be gained by the addition of this structure.
- 2) In the nucleotide binding pocket of the Irgb6 structure crystallized in the presence of GMPPNP, a density corresponding to the gamma-phosphate is lower than those of alpha- and beta- phosphates. There are several possible reasons for this, such as a partially hydrolyzed GMPPNP, or a mixture of GMPPNP and GDP, or the highly flexible nature of the gamma-phosphate, etc. In the current situation, therefore, we are not confident that this structure represents the GMPPNP form of Irgb6. We should further examine which type of biochemical state is represented by this crystal.

Thus, we have excluded the GMPPNP data from Figure S1C.

2. The modelling results combined with the loop-swap experiment give good evidence that Irgb6-PVM-loop equivalent in Irga6 cannot mediate phospholipid binding. I am interested as to whether the same is true of Irgb10. Could the authors conduct a similar loop-swap experiment in cells with the equivalent loop in Irgb10? I realise this may take a month or two if this mutant has not already been generated. As an alternative, could the authors use Glide modelling to see if the phospholipid could bind to Irgb10 in the same place? Presuming Irgb10 also cannot mediate phospholipid binding, this would drive home the unique nature of this phospholipid binding by Irgb6.

Thank you for the constructive comment. Since we did not make the Irgb10 mutant, we performed Glide docking of phospholipids using the Irgb10 structure. Surprisingly, several phospholipids docked to the pocket of Irgb10 in a similar manner to that observed in Irgb6. However, contrary to the PI5P preference observed in Irgb6, a preference for PS over other phospholipids was observed in Irgb10. Therefore, Irgb10 might utilize both the myristoylated glycine at the N-terminus and the phospholipids (especially PS) to bind PVM. We have included the docking results in Figure S4I-K with a discussion (page 17, lines 405-412).

Minor points:

1. The authors performed the lipid-binding modelling on the GTP-bound Irgb6 structure. Could the

authors clarify in the text whether they expect the lipid binding to occur in the nucleotide-free state of Irg6b? If they do not expect lipids to bind in this state, are there structural differences that support this?

We do not have direct structural evidence about whether the phospholipid binds to the pocket in the nucleotide free state. Since the membrane binding loop (α F- α Ga loop) was not visualized in the nucleotide free structure of Irg6b because of its high flexibility, we utilized the GTP-bound structure for the docking experiments. We clearly described this in the revised manuscript (page 11, lines 263-266).

2. Could the authors explicitly state in the text whether PI5P/PS binding is unique to Irg6b or is expected in other Irg family members?

We performed Glide docking of phospholipids to the Irga6 and Irgb10. Summary is listed below.

- ◆ Phospholipids are expected to bind to both Irga6 and Irgb10.
- ◆ PI5P preference is unique in Irgb6.
- ◆ PS preference was observed in Irgb10.
- ◆ Preference to PS or PE was observed in Irga6.

We included these results and discussions in the revised manuscript (Fig. S4G-K; page 17, lines 405-412).

3. The authors state that the G-domain of Irg6b-GTP is more similar to Irg6a-GDP compared to Irg6a-GMPPNP, giving the explanation that the Irg6b G2/G3 sequences do not coordinate the gamma phosphate, so overall the Irgb6-GTP G-domain recognizes the "GDP part" of the GTP. If one excludes the G2/G3 sequences of the Irg6a-GMPPNP structure, does it better align with the Irg6b-GTP structure, or is it still worse compared with the Irg6a-GDP structure?

We have aligned them without the G2/G3 sequence. Consequently, Irgb6-GTP was still better aligned with the Irga6-GDP than the Irga6-GMPPNP (RMSD 1.598 for Irga6-GDP; 2.169 for Irga6-GMPPNP). We described them in the revised manuscript (page 9, lines 209-210)

4. We thank the authors for providing the crystal structures. These were, on the whole, well built and

refined, particularly around the areas that the authors refer to in the manuscript. However there are a few regions where the model does not explain the density well, that should be addressed before release from the PDB. Irgb6-GTP: 145-148; 210-212; 359-362; 400-403. Irgb6-NF: 209-212.

We thank the reviewer for checking our structures. We have carefully refined both structures again.

5. Could the authors provide more detail on RMSD calculation in the methods? I am presuming, for example, they aligned the G domain of one structure to the G domain of another and reported the RMSD, rather than aligning the structure as a whole, then reporting the G domain RMSD, but this should be explicitly mentioned.

We have aligned them as suggested. We have included the method for RMSD calculation in the Materials and Methods of the revised manuscript (page 21, lines 480-484).

6. Could the authors clarify in the manuscript whether the Irgb6 rescue constructs in the cell experiments were overexpressed?

We retrovirally overexpressed the Irgb6 wild-type or the mutants. We clarified the procedure in the revised manuscript (page 14, lines 324-325).

7. What is the killing activity of wild type IFN- γ -stimulated MEFs for toxoplasma infection? Could the authors state this in the text for comparison?

We thank the reviewer for this question. We quantified luciferase-expressing *T. gondii* numbers by luciferase assays since the parasite numbers are proportional to the luciferase units (Yamamoto et al. *Immunity* 2012; Fig. S2D and S2E). *T. gondii* numbers in IFN- γ -stimulated cells relative to those in unstimulated cells was calculated by the luciferase units. To clarify the point, we rephrase the “killing activity” as “the IFN- γ -induced reduction of *T. gondii* numbers” in the revised manuscript (page 14, lines 355 and 328-329).

8. Figure 4A&B: the lipid molecules are not very obvious in this figure. Could the authors make them stand out more by, for example, making the sticks in each lipid molecule thicker?

We have revised the Figure 4A-B, as the reviewer suggested.

9. The authors state that the G domain in the GTP-bound structure is in a pre-isomerization state, referencing two papers describing an ATPase in a similar state. Are there any examples of other GTPases in this pre-isomerization state? Could authors include these?

We have included the references describing the isomerization step of small GTPases (page 7, lines 149-150).

A few parts of the manuscript require minor language clarification. Could the authors rephrase the following:

1. Page 2 Line 3: "it disrupts it". The usage of 'it' for two different objects is confusing.

We have revised it (page 2, line 27).

2. Page 2 Line 8-9: "It is recruited.... and disrupts it". The usage of 'it' for two different objects is confusing.

We have revised it (page 2, line 34).

3. Page 10 bottom paragraph: "Observed from the bottom side, the other side of the GTP-pocket...". The phrase "the other side of the GTP-pocket is confusing". Consider "the opposite face to the G domain binding face" or similar.

We have rephrased it as this reviewer suggested (page 10, line 236).

4. Page 10 bottom paragraph: "The C-domain helices rotate....". I'm presuming this is in the authors' structures, but it should be explicitly mentioned to avoid confusion.

We have revised it (page 10, line 242).

5. Page 12 second paragraph: "The Inositol makes...". Lower case "i" in "inositol".

We have revised it (page 12, line 285).

6. Page 15 Line 1: "Irgb6 has a crucial role to target the PVM...and destroy them". This implies that the Irgb6 destroys the PVM of *T. gondii*. If this is not the case could the authors rephrase this sentence?

We have revised it (page 15, line 347).

Reviewer #2 (Comments to the Authors (Required)):

The manuscript by Yumiko Saijo-Hamano and colleagues report the crystal structures of irgb6 from mouse. Irga6, irgb10 and irgb6 control intracellular Toxoplasma growth in interferon activated murine cells. Here authors solved the last structure of these major three ISGs, modeled the structure complexed with phospholipids and validated the model by point mutations, thus partially explained the PVM binding mechanism. The work is original and interesting but needs some revisions and serious proof-reading before publication.

Major findings include:

Authors solved the crystal structures of the last IRG among three major IRGs (irga6, irgb6 and irgb10) which all can be recruited on the PVM.

Modeling and cellular assays suggested a different structural basis of membrane recognition by irgb6, apart from previously reported irga6 and irgb10.

We thank the reviewer for deep understanding of our Irgb6 structure paper.

Major issues

Authors claims a GTP-bound structure in Fig 1. However, according to the density map in Fig S2a, I am not convinced the ligand is GTP. The author needs to provide the occupancy, the B factor for the ligand, as well as the Molprobity analysis for the ligand and the adjacent Mg²⁺. The twisted triphosphate tail seems unlikely as far as I can tell from Fig S2. Or is there a reasonable explanation about its orientation?

[Figure removed by editorial staff per authors' request]

We thank the reviewer for this constructive comment. As this reviewer suggested, we also think that the phosphate part of GTP in our structure seems more twisted than usual. During this revision stage, therefore, we carefully refined the Irgb6 structures, especially around the nucleotide-binding pocket.

The occupancies and the B-factors for the α -, β -, γ - Phosphates in refined GTP-bound structure are listed in the right panel. The B-factors of atoms in the γ -phosphate show relatively

higher values than those in the guanosine, but are as high as those in the α -, β - phosphates.

We also attached the validation report of the ligand below, which was generated during the PDB deposition. Although some outliers for bond angles, lengths, and torsions were still found, the electron density map corresponding to the GTP clearly shows the density corresponding to three phosphates. In addition, we also tried GDP instead of GTP, which resulted in a large residual density as well as an inappropriate position for β -phosphate. Therefore, we concluded that GTP should exist in the pocket but that it takes on a somewhat unusual geometry for phosphates. This twisted conformation might be caused by strain through excessive interactions between the three phosphates and P-loop and switch I residues (Fig. S2 B).

We have also checked the density for the Mg ion in the submitted manuscript because the position of this density was unusual for Mg and no coordinated waters were detected. Consequently, the water molecule was better fit to this density with the B-factor of 31.55 \AA^2 . Therefore, Mg was excluded from our PDB model, instead we replaced it with a water molecule.

[Figures removed by editorial staff per authors' request]

We have included these discussions in the revised manuscript ((page 5, lines 112-116; page 7, lines 152-154, Fig. S2)

I am not sure that the PI5P head is shown when looking at Fig 4a, especially in Fig 4d where only C4 ring can be clearly seen. Please make sure that the ligand is PI5P or adjust the panel for better visualization.

We have remade Figure 4 to clearly show the PI5P head (Inositol 1,5-bisphosphate (Ins(1,5)P₂)). We have also added the structures of phosphor-lipids we used (Fig. S4 A).

It would be nice to see simultaneous IFA staining for Irga6 or Irgb10 in Fig 5B. Do the mutations affect recruitments of other IRG proteins?

We thank this reviewer for the constructive comment. We have added the simultaneous staining for Irga6 and Irgb10, representing the mutations of Irgb6 significantly decreased the recruitments of Irga6 and Irgb10. These data have been included in the revised manuscript (page 14, lines 338-341, Fig. 5E-H).

It is mentioned that *Toxoplasma* inactivates IRGs by phosphorylation of the G loop SWI. This statement should also include the work from the Howard lab (PMID: 21203588) Since the G loop is described as being different in the current model, it would be important to discuss these prior findings in light of the new structure. Does the altered conformation of the G loop make this region more or less accessible? Can the authors comment on how PO₄ might alter the conformation to either prevent binding of GTP or hydrolysis?

We are sorry for failing to include the appropriate reference. Since the threonine residues (T89 and T95) in our structures show high B-factors and residues 90-94 are invisible because of their high flexibilities, we could not mention their structural details. Therefore, we only note the biological importance of these threonine residues in the discussion section. In the revised manuscript, we have added the reference as this reviewer suggested (page 18, line 421).

Minor issues

In Fig1A, cyan was used to color N-domain but also for the general GTP-bound form. Cyan was also

used to color G-domain in Fig1C, which was colored as green in Fig1A. I suggest recoloring N-domain in Fig1A to avoid confusing.

We have changed the coloring of Figure 1A as suggested.

The completeness of the nucleotide-free data is low (especially when comparing with other two datasets) in the last shell. The reflection of a nominal high resolution here has a comparatively big difference between the R-work and R-free, especially in terms of the 1.8Å resolution. Authors might need to be more conservative in estimating the resolution to create a map with less noise and more accuracy.

Thank you for the constructive comment. We have re-refined the nucleotide-free data and set the high resolution limit for refinement at 2.0 Å (Table S1).

In the last paragraph of the results (very inconvenient because of lacking line numbers), authors are referring Fig 5b that is describing irgb6 recruitment. Fig 5b and 5d need to be changed reciprocally according to authors description in the paper.

We have changed the order of panels in Fig. 5 as suggested.

There is no Fig S4 while it was referred in the 'in situ docking stimulation' section.

No Fig S5 as well.

Ghosh, Molecular Cell, 2004, reference duplicated.

We are sorry for the errors and corrected them in the revised manuscript.

Reviewer #3 (Comments to the Authors (Required)):

This is a timely and well written structure-function report that informs the mechanism of IRGb6-membrane interaction. The study is important to our understanding of cell autonomous immune sensing for a range of vacuole-resident pathogens. In this way, it is appropriate for the readership of LSA. IRGb6 belongs to the family of p47 GTPases. Other members of this family, namely IRGa6 and IRGb10, have had their crystal structures solved. These proteins use a myristoylated glycine to interact with target membranes. However, this motif is not conserved in the paralog IRGb6. Instead, the C-terminal amphipathic domain has been shown to mediate phospholipid interactions, however, the structural mechanism of membrane binding remains is not known. This paper addresses this information gap by solving the first structures of IRGb6 in the GTP (1.5 Angstrom) bound and unbound state (1.9 Angstrom).

We thank this reviewer for deep understanding of our Irgb6 structure paper.

The central findings of the paper are that the G-domain, which contains the nucleotide binding pocket, undergoes a similar catalytic shift to other p47 IRGs based on structure comparison. This is consistent with the existing model where GTPase activity regulates homo-dimerization. In contrast to IRGb10 and IRGa6, the C-terminal domain of IRGb6, containing the alphaF-alphaGa "PVM binding loop," also undergoes a conformational shift in the GTP bound state. The GTP bound structure was used to model the interaction between the alphaF-alphaGa loops and phospholipids. PI5P was predicted to have the lowest binding free energy, suggesting this is a preferential interaction over PS, PE or PC. These results (Figure 4) somewhat recapitulate PIP strip data from (Lee et al 2019), however, in that study PE and PC binding were not observed, which could be better discussed.

Thank you for following our previous LSA paper in which the PIP strip showed the strongest binding of PI5P, the moderate binding of PS, and no binding of PE and PC. Glide docking scores consistently showed the best fitting of PI5P head, whereas the scores for PS head and PE head did not show significant differences. In addition to the docking score, we revised figure 4A and 4B to visualize clearly the fitting of phospho-lipids to the pocket. Tip phosphates of all phospho-lipids interact with R371, and L279 supports all phospho-lipid binding through hydrophobic contacts. PI5P then extends toward the hydrophobic cluster (right side of figure 4A), PS partially extends toward the same direction (right side of figure 4B), but PE and PC only occupy the left side of the pocket. Hence, the acyl chain of PI5P is expected to bind tightly to the pocket, that of PS will bind moderately and those of PE and PC will not. These properties

for acyl chain binding are consistent well with Fig. S4E and F. In the revised manuscript, these discussions are included (page 12, lines 283-292).

To evaluate the significance of the PVM binding loop, a chimera between IRGb6 and IRGa6 was generated where the region between 277-286 was swapped, the 3 glycine residues were mutated, or the N-terminus tryptophan was substituted with alanine. The conclusion was that the glycine residues are necessary for optimal PVM localization and parasite restriction. However, additional controls seem important to show that the introduced point mutations in IRGb6 specifically inhibit phospholipid binding. For example, do these constructs lose PIP5 and PS binding specificity using PIP strips or a related approach (Lee et al 2020)?

About the specificity of PI5P, Glide docking of phospho-lipids to the 3 glycine mutant and W3A mutant was performed. Although PI5P is still preferred over other phospholipids in the 3 glycine mutant, the mean Glide Scores for PI5P-mutant are worse than for PI5P-Irgb6. For the W3A mutant, the docking scores became worse for every phospholipid and the preference for PI5P was lost. Thus, docking experiments to the mutants further support the specificity of PI5P to the Irgb6 pocket. We have included these discussions in the revised manuscript (page 13, lines 353-361).

A related concern is that it is possible that loss of PVM localization/parasite restriction is simply due to misfolding of the mutant constructs in the ER. A co-stain for ER markers would help limit this concern and strengthen a model wherein these amino acids are essential for PVM localization.

To answer the reviewer's concern, the localization of Irgb6 mutants was examined in the reconstituted MEFs by indirect immunofluorescent assay. In uninfected cells, Flag-tagged Irgb6_a6(all), the G277D/G285T/G286F, or the W3A mutants was localized at cytoplasm in a manner similar to Flag-tagged wild-type Irgb6 (Fig. S5 A). Furthermore, we found that the Irgb6 mutants as well as wild-type Irgb6 were not completely co-localized with ER (Fig. S5 A), suggesting that the defective localization of the Irgb6 mutants on *T. gondii* PV may not be due to misfolding of these proteins on ER.

Minor comments

-In Figure 5, a schematic, showing the position of point mutations along the protein would be useful to the reader.

We added a schematic presentation in revised figure 5A.

-Including a statistic regarding the number of vacuoles and parasites evaluated would be useful to assess the robustness of the data presented in 5C-D.

We numbered nearly one hundred of GRA2-positive *T. gondii* as PV in one experiment and performed three experiments. The original data were shown in figure S5B.

[Figure removed by editorial staff per authors' request]

Regarding figure 5D, we did not directly number *T. gondii* numbers but indirectly quantified luciferase-expressing *T. gondii* numbers by luciferase assays since the parasite numbers are proportional to the luciferase units (Yamamoto et al. *Immunity* 2012; Fig. S2D and S2E). *T. gondii* numbers in IFN- γ -stimulated cells relative to those in unstimulated cells was calculated by the luciferase units and shown in Fig. 5D. The indirect count of *T. gondii* numbers is now clarified in Materials and Methods (page 24, lines 570-571).

October 5, 2021

Re: Life Science Alliance manuscript #LSA-2021-01149-TR

Prof. Ryo Nitta
Kobe University Graduate School of Medicine
Division of Structural Medicine and Anatomy
7-5-1 Kusunoki-cho
Chuo-ku
Kobe, Hyogo 650-0017
Japan

Dear Dr. Nitta,

Thank you for submitting your revised manuscript entitled "Structural basis of membrane recognition of *Toxoplasma gondii* vacuole by Irgb6" to Life Science Alliance. The manuscript has been seen by the original reviewers whose comments are appended below. While reviewers 2 and 3 are now completely satisfied about the work in terms of its suitability for Life Science Alliance, few important issues remain for Reviewer 1 to be addressed.

Our general policy is that papers are considered through only one revision cycle; however, given that the suggested changes are relatively minor, we are open to one additional short round of revision. Please note that I will expect to make a final decision without additional reviewer input upon resubmission.

Please submit the final revision within one month, along with a letter that includes a point by point response to the remaining reviewer comments.

To upload the revised version of your manuscript, please log in to your account: <https://lsa.msubmit.net/cgi-bin/main.plex>
You will be guided to complete the submission of your revised manuscript and to fill in all necessary information.

B. MANUSCRIPT ORGANIZATION AND FORMATTING:

Sincerely,

Reviewer #1 (Comments to the Authors (Required)):

The authors have addressed the majority of my comments well. In particular Figure 4 now looks very nice, with the lipid moiety obvious to the reader. However there are a couple of outstanding issues that need to be addressed before publication. However once these issues are addressed, this manuscript will be ready for publication.

From previous review:

"7. What is the killing activity of wild type IFN-gamma-stimulated MEFs for toxoplasma infection? Could the authors state this in the text for comparison?"

"We thank the reviewer for this question. We quantified luciferase-expressing *T. gondii* numbers by luciferase assays since the parasite numbers are proportional to the luciferase units (Yamamoto et al. Immunity 2012; Fig. S2D and S2E). *T. gondii* numbers in IFN- γ -stimulated cells relative to those in unstimulated cells was calculated by the luciferase units. To clarify the point, we rephrase the "killing activity" as "the IFN- γ -induced reduction of *T. gondii* numbers" in the revised manuscript (page 14, lines 355 and 328-329)."

I am afraid the authors may have misunderstood my question - apologies for this. I would like to know what the toxoplasma survival rate is in wild-type MEF cells stimulated with IFN-gamma (rather than in *Irgb6*-KO cells). This is important to contextualise the effect of the knockout and the subsequent reconstitutions. Could the authors state this rate in the manuscript for comparison?

Minor formatting/wording issues:

1. The first line of the discussion (line 354, page 16) still does not make sense to me, as it suggests that *Irgb6* destroys the PVM - is this true, or is it destroyed by other proteins? If the latter option is true, consider "*Irgb6* has a crucial role to target the PVM of *T. gondii* to facilitate its destruction".
2. The order of the figure legends for Fig 5 now does not match the figure. Can the authors correct this?
3. Also could the authors include a "*Irgb6* k/o MEF" label somewhere in Figure 5? Upon re-reading it took me a while to remember that figure 5 is in an *Irgb6* k/o line. This is a minor issue as the label is in the legend.
4. Figure 6 has two panels - A and B which need explicit legends.
5. Figure S4A: The figures show the structures of PI5P, etc., rather than the structural formulae.
6. Many of the symbols are not rendered correctly in the pdf versions of the figures (e.g. the alpha symbols in Fig 1A). They are rendered in the powerpoint source files, so presumably something has gone wrong in the conversion process.

Reviewer #2 (Comments to the Authors (Required)):

Thank you for the careful attention to my earlier comments. I find the responses adequately address my questions and I have no further concerns.

Reviewer #3 (Comments to the Authors (Required)):

This is a timely and well-written structure-function report that informs the mechanism of *IRGb6*-membrane interaction. The study is important to our understanding of cell-autonomous immune sensing for a range of vacuole-resident pathogens. The authors adequately responded to the comments of all reviewers and included new data glide modeling with the *Irgb10* phospholipid-binding loop. There is a significant modification to the discussion and better description of the parasite growth assay and immunofluorescence experiments. Overall this is an improved manuscript within the scope of LSA.

Reviewer #1 (Comments to the Authors (Required)):

The authors have addressed the majority of my comments well. In particular Figure 4 now looks very nice, with the lipid moiety obvious to the reader. However there are a couple of outstanding issues that need to be addressed before publication. However once these issues are addressed, this manuscript will be ready for publication.

Firstly, we are deeply grateful for the kind, detailed review by this reviewer.

From previous review:

"7. What is the killing activity of wild type IFN-gamma-stimulated MEFs for toxoplasma infection? Could the authors state this in the text for comparison?"

"We thank the reviewer for this question. We quantified luciferase-expressing *T. gondii* numbers by luciferase assays since the parasite numbers are proportional to the luciferase units (Yamamoto et al. Immunity 2012; Fig. S2D and S2E). *T. gondii* numbers in IFN- γ -stimulated cells relative to those in unstimulated cells was calculated by the luciferase units. To clarify the point, we rephrase the "killing activity" as "the IFN- γ -induced reduction of *T. gondii* numbers" in the revised manuscript (page 14, lines 355 and 328-329)."

I am afraid the authors may have misunderstood my question - apologies for this. I would like to know what the toxoplasma survival rate is in wild-type MEF cells stimulated with IFN-gamma (rather than in *Irgb6*-KO cells). This is important to contextualise the effect of the knockout and the subsequent reconstitutions. Could the authors state this rate in the manuscript for comparison?

We are sorry for misunderstanding your comment. As you have already realized, the wild-type MEF cells were not assigned as a control in this experimental design. Using the *Irgb6*-KO MEFs, transfection of the empty vector served as a negative control (without *Irgb6*), whereas that of the wild type *Irgb6* served as a positive control (with *Irgb6*). In comparison with these controls, the killing activities of several structure-based *Irgb6* mutants were examined.

In addition to this experimental design, we previously reported that the survival rates of *T. gondii* in wild-type MEF were around 10-20 % (<https://www.life-science-alliance.org/content/4/7/e202000960>, Figs. 2C, 4E, 6C). Hence, we added the comparison with the wild type MEFs by adding the following sentence in the revised manuscript with the reference of this paper (p.14 lines 337-340).

=====

Reconstitution of wild-type Irgb6 in Irgb6 KO MEFs resulted in almost 20% *T. gondii* survival, which was largely similar to the parasite survival in IFN- γ -stimulated wild-type MEFs (Pradipta et al. 2021).

=====

Minor formatting/wording issues:

1. The first line of the discussion (line 354, page 16) still does not make sense to me, as it suggests that Irgb6 destroys the PVM - is this true, or is it destroyed by other proteins? If the latter option is true, consider "Irgb6 has a crucial role to target the PVM of *T. gondii* to facilitate its destruction".

Thank you for the constructive advice. The latter option makes sense. Therefore, we have revised this sentence as this reviewer suggested. (p. 16, line 350)

2. The order of the figure legends for Fig 5 now does not match the figure. Can the authors correct this?

We have corrected in the revised manuscript. (p.32, legend of figure 5)

3. Also could the authors include a "Irgb6 k/o MEF" label somewhere in Figure 5? Upon re-reading it took me a while to remember that figure 5 is in an Irgb6 k/o line. This is a minor issue as the label is in the legend.

We have included the "Irgb6 k/o MEF" label in the panel A. (Fig. 5A)

4. Figure 6 has two panels - A and B which need explicit legends.

We have added the legend. (p.32, legend of figure 6)

5. Figure S4A: The figures show the structures of PI5P, etc., rather than the structural formulae.

We have corrected it as this reviewer suggested. (p.33, legend of figure S4A)

6. Many of the symbols are not rendered correctly in the pdf versions of the figures (e.g. the alpha symbols in Fig 1A). They are rendered in the powerpoint source files, so presumably something has gone wrong in the conversion process.

We uploaded the PDF version of figures in which we could not find any rendering errors.

October 11, 2021

RE: Life Science Alliance Manuscript #LSA-2021-01149-TRR

Prof. Ryo Nitta
Kobe University Graduate School of Medicine
Division of Structural Medicine and Anatomy
7-5-1 Kusunoki-cho
Chuo-ku
Kobe, Hyogo 650-0017
Japan

Dear Dr. Nitta,

Thank you for submitting your revised manuscript entitled "Structural basis of membrane recognition of *Toxoplasma gondii* vacuole by Irgb6". We would be happy to publish your paper in Life Science Alliance pending final revisions necessary to meet our formatting guidelines.

- please add a Category for your manuscript in our system
- please add the Twitter handle of your host institute/organization as well as your own or/and one of the authors in our system
- please add a conflict of interest statement to your main manuscript text
- please use the [10 author names, et al.] format in your references (i.e. limit the author names to the first 10)
- please revise the figure legend for figure 6 such that the figure panels are introduced in an alphabetical order
- please add a callout for Figure S6C to your main manuscript text

Open questions:

- there is no approval statement and no separate Data Availability section

A. FINAL FILES:

B. MANUSCRIPT ORGANIZATION AND FORMATTING:

Sincerely,

October 18, 2021

RE: Life Science Alliance Manuscript #LSA-2021-01149-TRRR

Prof. Ryo Nitta
Kobe University Graduate School of Medicine
Division of Structural Medicine and Anatomy
7-5-1 Kusunoki-cho
Chuo-ku
Kobe, Hyogo 650-0017
Japan

Dear Dr. Nitta,

Thank you for submitting your Research Article entitled "Structural basis of membrane recognition of *Toxoplasma gondii* vacuole by Irgb6". It is a pleasure to let you know that your manuscript is now accepted for publication in Life Science Alliance. Congratulations on this interesting work.

DISTRIBUTION OF MATERIALS:

Again, congratulations on a very nice paper. I hope you found the review process to be constructive and are pleased with how the manuscript was handled editorially. We look forward to future exciting submissions from your lab.

Sincerely,
